# MOVE: Unsupervised Movable Object Segmentation and Detection

**Adam Bielski**
University of Bern
adam.bielski@unibe.ch

**Paolo Favaro**
University of Bern
paolo.favaro@unibe.ch

## Abstract

We introduce MOVE, a novel method to segment objects without any form of supervision. MOVE exploits the fact that foreground objects can be shifted locally relative to their initial position and result in realistic (undistorted) new images. This property allows us to train a segmentation model on a dataset of images without annotation and to achieve state of the art (SotA) performance on several evaluation datasets for unsupervised salient object detection and segmentation. In unsupervised single object discovery, MOVE gives an average CorLoc improvement of 7.2% over the SotA, and in unsupervised class-agnostic object detection it gives a relative AP improvement of 53% on average. Our approach is built on top of self-supervised features (e.g. from DINO or MAE), an inpainting network (based on the Masked AutoEncoder) and adversarial training.

## 1 Introduction

Image segmentation and object detection are today mature and essential components in vision-based systems with applications in a wide range of fields including automotive [1], agriculture [2], and medicine [3], just to name a few. A major challenge in building and deploying such components at scale is that they require costly and time-consuming human annotation. This has motivated efforts in self-supervised learning (SSL) [4–6]. The aim of SSL is to learn general-purpose image representations from large unlabeled datasets that can be fine-tuned to different downstream tasks with small annotated datasets. While SSL methods have also been fine-tuned for image segmentation since their inception, it is only with the recent state of the art (SotA) methods, such as DINO [4] and Dense Contrastive Learning [6], that a clear and strong link to object segmentation has been observed. This has led to several methods for salient object detection built on top of SSL features [7–12].

Most prior work based on SSL features defines some form of clustering by either using attention maps [7–9] or similarity graphs [10–12]. In this work, we take a quite different direction. Rather than directly clustering the features, we train a network to map them to a segmentation mask. As supervision signal we use the *movability* of objects, *i.e.*, whether they can be locally shifted in a realistic manner. We call our method MOVE. This property holds for objects in the foreground, as they occlude all other objects in the scene. This basic idea has already been exploited in prior work with relative success [13–19]. Nonetheless, here we introduce a novel formulation based on movability that yields a significant performance boost across several datasets for salient object detection.

In our approach, it is not necessary to move objects far from their initial location or to other images [14, 16] and thus we do not have to handle the context mismatch. It is also not necessary to employ models to generate entire scenes [15, 20], which can be challenging to train. Our working principle exploits observations also made by [17–19]. They point out that the correct mask maximizes the inpainting error both for the background and the foreground. However, rather than using the

36th Conference on Neural Information Processing Systems (NeurIPS 2022).

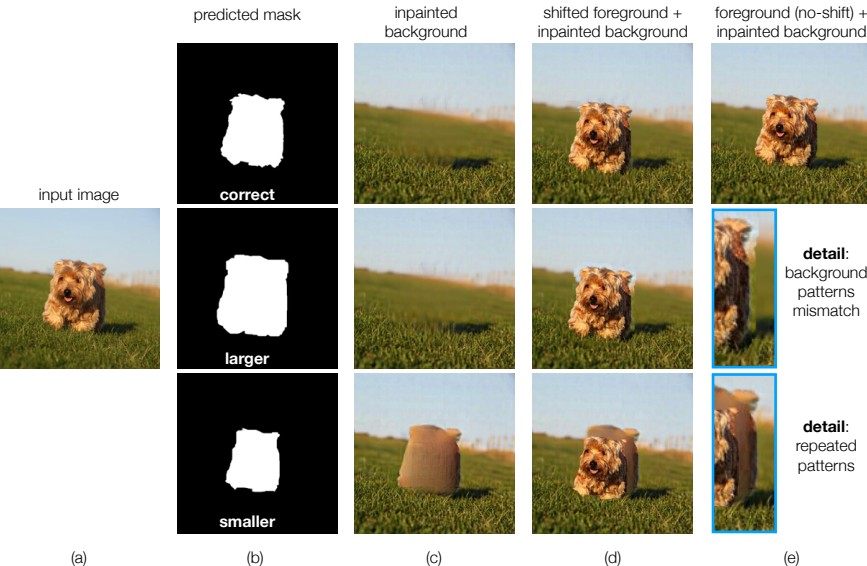

Figure 1: Exploiting inpainting and movability. (a) Input image. (b) Examples of predicted segmentation masks: correct (top), larger (middle) and smaller (bottom). (c) Inpainted backgrounds in the three corresponding cases. (d) Composite image obtained by shifting the foreground object in the three cases. (e) It can be observed that when the mask is incorrect (it includes parts of the background or it does not include all of the background), the background inpainting combined with shifting reveals repeated patterns and mismatching background texture, when compared to the original input image or composite images obtained without shifting.

reconstruction error as a supervision signal, we rely on the detection of artifacts generated through shifting, which we find to provide a stronger guidance.

Suppose that, given a single image (Figure 1 (a)), we predict a segmentation mask (one of the 3 cases in Figure 1 (b)). With the mask we can remove the object and inpaint the background (Figure 1 (c)). Then, we can also extract the foreground object, randomly shift it locally, and paste it on top of the inpainted background (Figure 1 (d)). When the mask does not accurately follow the outline of a foreground object (e.g., as in the middle and bottom rows in Figure 1), we can see duplication artifacts (of the foreground or of the background). We exploit these artifacts as supervision signal to detect the correct segmentation mask. As inpainter we use a publicly available Masked AutoEncoder (MAE) [21] trained with an adversarial loss.[1] Our segmenter uses a pre-trained SSL ViT as backbone (e.g., DINO [4] or the MAE encoder [21]). We then train a neural network head based on an upsampling Convolutional Neural Network (CNN). Following [12], we also further refine the segmenter by training a second segmentation network (SelfMask [12]) with supervision from pseudo-masks generated by our trained segmenter. Even without these further refinements MOVE shows a remarkable performance on a wide range of datasets and tasks. In particular, in unsupervised single object discovery on VOC07, VOC12 and COCO20K it improves the SotA CorLoc between 6.1% and 9.3%, and in unsupervised class agnostic object detection on COCOval2017 it improves the $AP_{50}$ by 6.8% (a relative improvement of 56%), the $AP_{75}$ by 2.3% (relative 55%) and the AP by 2.7% (relative 49%).

## 2   Method

Our objective is to train a segmenter to map a real image $x \in \mathbb{R}^{H \times W \times 3}$, with $H$ the height and $W$ the width of the image, to a mask $m \in \mathbb{R}^{H \times W}$ of the foreground, such that we can synthesize a realistic image for any small shifts of the foreground. The mask allows to cut out the foreground from $x$ and to move it arbitrarily by some $\delta \in \mathbb{R}^2$ shift (see Figure 2, top-left). However, when the shifted foreground is copied back onto the background, missing pixels remain exposed. Thus, we

---

[1]`https://github.com/facebookresearch/mae/blob/main/demo/mae_visualize.ipynb`

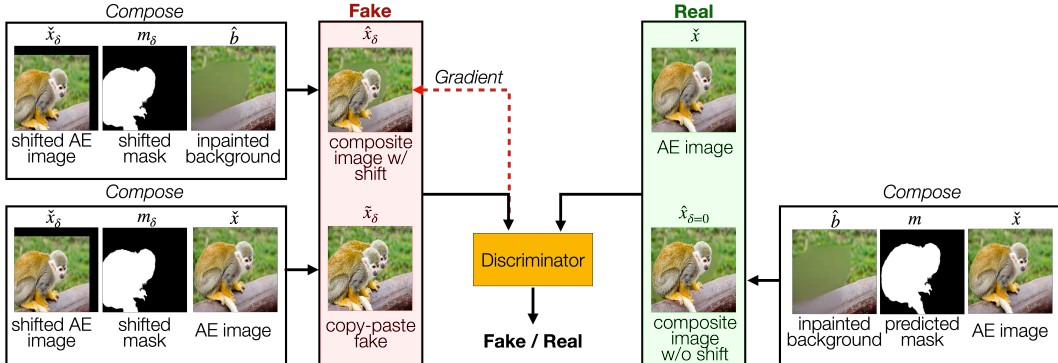

Figure 2: Synthetic and real images used to learn how to segment foreground objects. We obtain the predicted mask and inpainted background from our segmenter and MAE respectively. We train the segmenter in an adversarial manner so that the composite image with a shifted foreground (left, top row) looks real. A discriminator is trained to distinguish two types of real (right) from two types of fake (left) images. The fake images consist of the composite image with a shift and a copy-paste image, obtained by placing the shifted foreground on top of the input image. The set of real images consists of composite images without a shift and the real images. The real images are first autoencoded with MAE to match the artifacts of the inpainted background.

inpaint the background with a *frozen* pre-trained MAE[2] and obtain $\hat{b}$ (see Figure 3). Moreover, there is a difference between the texture of $\hat{b}$, which is generated from a neural network, and the texture of the cut out foreground from $x$, which is a real image. To ensure more similarity between these two textures, we synthesize $\hat{x}_\delta$ by extracting the foreground from the autoencoding (AE) of the input image $x$ shifted by $\delta$, which we call $\check{x}_\delta$, and by pasting it onto the background $\hat{b}$.

We enforce the realism of the synthesized images $\hat{x}_\delta$ by using adversarial training, i.e., by training the segmenter against a discriminator that distinguishes two sets of *real* (Figure 2, right hand side) from two sets of *fake* images (Figure 2 left hand side). The synthetic *real* image $\hat{x}_{\delta=0}$ is obtained by composing a zero-shifted foreground with the inpainted background; the second *real* image $\check{x}$ is instead simply the AE of $x$. The two *fake* images are obtained by composing a $\delta$-shifted foreground with either the inpainted background $\hat{b}$ or $\check{x}$, and obtain $\hat{x}_\delta$ and $\tilde{x}_\delta$ respectively.

We introduce all the above synthetic images so that the discriminator pays attention only to artifacts due to incorrect masks from the segmenter. Ideally, the segmenter should generate masks such that the fake image $\hat{x}_\delta$ looks as realistic as $\check{x}$ for any small $\delta$. However, the discriminator might distinguish these two images because of the background inpainting artifacts and not because of the artifacts due to an incorrect segmentation (which are exposed by random shifts). To avoid this undesired behavior, we also introduce the real image $\hat{x}_{\delta=0}$. This image has no segmentation artifacts for any mask, but has the same background inpanting artifacts as the fake images (although there is no shift in $\hat{x}_{\delta=0}$, the background inpainting creates artifacts beyond the boundaries of the segmentation mask). Finally, to guide the discriminator to detect repeated patterns (as those caused by incorrect masks, see Figure 1), we also add a fake image $\tilde{x}_\delta$, where the background has the original foreground.

The segmenter is trained only through the backpropagation from $\hat{x}_\delta$. The details of the segmentation network, the inpainting network and the adversarial training are explained in the following sections.

## 2.1 Segmenter

Following the recent trend of methods for unsupervised object segmentation [7–12, 22], we build our method on top of SSL features, and, in particular, DINO [4] or MAE [21] features. Thus, as a backbone, we adopt the Vision Transformer (ViT) architecture [23]. Following the notation in [10], we split an image $x \in \mathbb{R}^{H \times W \times 3}$ in tiles of size $P \times P$ pixels, for a total of $N = HW/P^2$ tiles (and we assume that $H$ and $W$ are such that $H/P$ and $W/P$ are integers). Each tile is then mapped

---

[2]The MAE [21] we use is based on a ViT architecture and has been pre-trained in an adversarial fashion (as opposed to the standard training with an MSE loss) to output more realistic-looking details

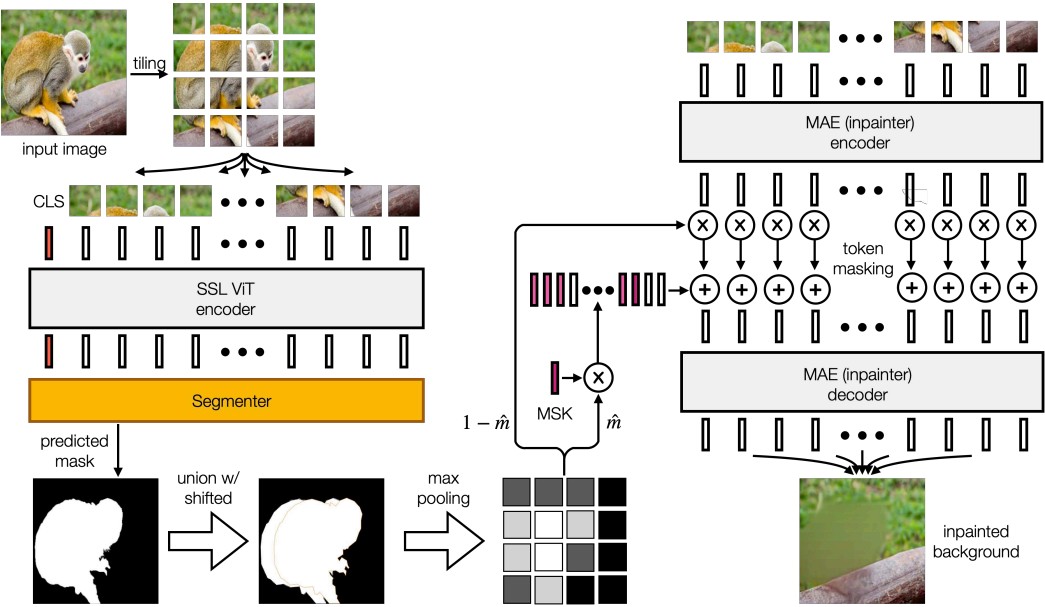

Figure 3: (Left) The segmenter is built on top of SSL features from a *frozen* encoder. To define the inpainting region for the background, the predicted mask is shifted and combined with the unshifted mask (bottom left). For better visualization purposes we highlight the edge of the shifted mask, but this does not appear in the actual union of the masks. This mask union is then downsampled to the size of the tile grid via max pooling and denoted $\hat{m}$. (Right) The inpainter is based on a *frozen* MAE. First, it takes all the tiles from the input image and feeds them to the MAE encoder. Second, it takes a convex combination between the encoder embeddings and the MSK learned embedding (but now frozen), where the convex combination coefficients are based on the downsampled mask $\hat{m}$. Finally, this combination is fed to the MAE decoder to generate the inpainted background.

through a trainable linear layer to an embedding of size $d$ and an additional CLS token is included in the input set (see Figure 3 left).

The *segmenter* network is a CNN that takes SSL features as input (e.g., from a pre-trained DINO or MAE encoder), upsamples them and then outputs a mask for the original input image. The final output is generated by using a sigmoid to ensure that the mask values are always between $0$ and $1$. We also ensure a minimum size of the support of the predicted mask by using

$$\mathcal{L}_{\min} = \frac{1}{n} \sum_{i=1}^{n} \max \left\{ \theta_{\min} - \sum_p \frac{m^{(i)}[p]}{HW}, 0 \right\} \tag{1}$$

where $n$ is the number of images in the training dataset, $m^{(i)}$ is the predicted mask from image $x^{(i)}$, $p$ is a pixel location within the image domain, and $\theta_{\min}$ is a threshold for the minimum mask coverage percentage respectively (in the range $[0, 1]$, where $0$ implies that the mask is empty and $1$ implies that the mask covers the whole image). Since masks should only take binary values to clearly indicate a segment, we use a loss that encourages $m^{(i)}$ to take either $0$ or $1$ values

$$\mathcal{L}_{\text{bin}} = \frac{1}{n} \sum_{i=1}^{n} \frac{1}{HW} \sum_p \min \left\{ m^{(i)}[p], 1 - m^{(i)}[p] \right\}. \tag{2}$$

## 2.2 Differentiable inpainting

The main task of MOVE is to predict a segmentation mask that can be used to synthesize a realistic image, where the foreground object is shifted on top of the inpainted background (see Figure 1 (e) top and Figure 2 top left). Figure 3 shows how we use the predicted high resolution mask for inpainting with MAE. Since MAE performs inpainting by masking or retaining entire patches of $P' \times P'$ pixels, it is necessary to also split the segmentation mask into a grid of tiles of $P' \times P'$ pixels and to map each tile to a single scalar between $0$ and $1$. We do that by using a max pooling operation within each tile and obtain a low-res mask $\hat{m}$, such that $1 - \hat{m}$ does not contain any part of the predicted mask. To regularize the predicted mask $m$, the mask losses $\mathcal{L}_{\min}, \mathcal{L}_{\text{bin}}$ are also computed on max

pool $\hat{m}$ and average pool downsampled masks (at a scale $1/P'$ of the original image resolution; for more details see the supplementary material). Then, we feed the entire set of image tiles to the MAE encoder and obtain embeddings $\xi_1, \ldots, \xi_N$. Next, for $j = 1, \ldots, N$, we compute the convex combination between the embeddings $\xi_j$ and the learned MSK (masked) token from MAE by using the low res mask $\hat{m}$ as $\hat{\xi}_j = \hat{m}[j] \cdot \xi_{\text{MSK}} + (1 - \hat{m}[j]) \cdot \xi_j$. Finally, we feed the new embeddings $\hat{\xi}_j$ in the MAE decoder and reassemble the output tiles back into the inpainted background image $\hat{b}$ (see Figure 3 bottom-right). Notice that we feed all the tiles as input to obtain a differentiable mapping that we can backpropagate on. Interestingly, we found that when no tile is masked at the input of the MAE encoder, the embeddings $\xi_j$ do not store significant information about their neighbors (see the supplementary material). This is in contrast to the typical use of MAE, where only the subset of "visible" tiles is fed as input to the encoder. However, such tile selection operation would make the inpainting not differentiable.

## 2.3 Adversarial training

Figure 2 shows how we create the images used in the adversarial training. First, we mask the input image with the predicted mask and compose with the inpainted background image, obtaining

$$\hat{x}_\delta[p] = m_\delta[p]\check{x}[p + \delta] + (1 - m_\delta[p])\hat{b}[p], \tag{3}$$

where $m_\delta[p] = m[p + \delta]$, $\delta \in [-\Delta W, \Delta W] \times [-\Delta H, \Delta H]$ is a 2D shift, with $\Delta$ the maximum shift range (relative to the image size). To make the inpainting artifacts in the no-shift composite image $\hat{x}_{\delta=0}$ more comparable to those in the shifted composite image, we define the background inpainting region as the union between the predicted mask and its shifted version (see Figure 3). Thus,

$$\hat{m} = \text{maxpool}_P(1 - (1 - m) \odot (1 - m_\delta)). \tag{4}$$

To improve the discriminator's ability to focus on repeated patterns artifacts, we additionally create *fake* images with a predicted shifted foreground pasted on top of the autoencoded image, obtaining $\tilde{x}_\delta = \check{x}_\delta \odot m_\delta + \check{x} \odot (1 - m_\delta)$.

The adversarial loss for the discriminator can be written as

$$\mathcal{L}_{\text{advD}} = -\mathbb{E}_{x_R} \min\{0, D(x_R) - 1\} - \mathbb{E}_{x_S} \min\{0, -D(x_S) - 1\} \tag{5}$$

where samples for "real" images $x_R$ are the set $\{\tilde{x}^{(i)}\}_{i=1,\ldots,n} \bigcup \{\hat{x}_{\delta=0}^{(i)}\}_{i=1,\ldots,n}$ and samples for synthetic images $x_S$ are the set $\{\hat{x}_\delta^{(i)}\}_{i=1,\ldots,n} \bigcup \{\tilde{x}_\delta^{(i)}\}_{i=1,\ldots,n}$, with uniform random samples $\delta \sim \mathcal{U}_2([-\Delta W, \Delta W] \times [-\Delta H, \Delta H])$ and $\mathbb{E}$ denotes the expectation. To speed up the convergence, we also use the projected discriminator method [24]. For the segmenter, we use instead the standard loss computed on the composite shifted images

$$\mathcal{L}_{\text{advS}} = -\mathbb{E}_{\hat{x}_\delta} D(\hat{x}_\delta). \tag{6}$$

Finally, with $\lambda_{\text{min}}, \lambda_{\text{bin}}$ nonnegative hyperparameters, our optimization is the adversarial minimization

$$S^* = \arg\min_S \mathcal{L}_{\text{advS}} + \lambda_{\text{min}}\mathcal{L}_{\text{min}} + \lambda_{\text{bin}}\mathcal{L}_{\text{bin}} \tag{7}$$

$$\text{subject to } D^* = \arg\min_D \mathcal{L}_{\text{advD}}. \tag{8}$$

## 3 Implementation

Except for the ablation studies, in all our experiments we use a self-supervised DINO [4] ViT-S/8 transformer pre-trained on ImageNet [25] as an SSL feature extractor. We take the output of the penultimate transformer block of DINO as the feature tokens with $P = 8$ and feed them to the segmenter. Our segmenter is a small upsampling convolutional neural network. It assembles the DINO features into a grid of size $H/P \times W/P$ and processes them with 3 upsampling blocks, so that the output matches the input image resolution. Each upsampling block first performs a $2 \times 2$ nearest upsampling, followed by a $3 \times 3$ convolutional layer with padding, batch normalization [26] and a LeakyReLU activation function (see the supplementary material for details). We add an additional block without upsampling and a linear projection to 1 channel, representing the mask. Our inpainting network is a ViT-L/16 transformer pre-trained on ImageNet as a Masked Autoencoder

Table 1: Comparison to the unsupervised saliency detection methods on 3 benchmarks

| Model | DUT-OMRON [29] | | | DUTS-TE [30] | | | ECSSD [31] | | |
|---|---|---|---|---|---|---|---|---|---|
| | Acc | IoU | max$F_\beta$ | Acc | IoU | max$F_\beta$ | Acc | IoU | max$F_\beta$ |
| HS [32] | .843 | .433 | .561 | .826 | .369 | .504 | .847 | .508 | .673 |
| wCtr [33] | .838 | .416 | .541 | .835 | .392 | .522 | .862 | .517 | .684 |
| WSC [34] | .865 | .387 | .523 | .862 | .384 | .528 | .852 | .498 | .683 |
| DeepUSPS [35] | .779 | .305 | .414 | .773 | .305 | .425 | .795 | .440 | .584 |
| SelfMask pseudo* [12] | .811 | .403 | - | .845 | .466 | - | .893 | .646 | - |
| BigBiGAN [36] | .856 | .453 | .549 | .878 | .498 | .608 | .899 | .672 | .782 |
| E-BigBiGAN [36] | .860 | .464 | .563 | .882 | .511 | .624 | .906 | .684 | .797 |
| Melas-Kyriazi et al. [37] | .883 | .509 | - | .893 | .528 | - | .915 | .713 | - |
| LOST [10] | .797 | .410 | .473 | .871 | .518 | .611 | .895 | .654 | .758 |
| Deep Spectral [22] | - | .567 | - | - | .514 | - | - | .733 | - |
| TokenCut [11] | .880 | .533 | .600 | .903 | .576 | .672 | .918 | .712 | .803 |
| FreeSOLO [8] | .909 | .560 | .684 | .924 | .613 | .750 | .917 | .703 | .858 |
| **MOVE (Ours)** | **.923** | **.615** | **.712** | **.950** | **.713** | **.815** | **.954** | **.830** | **.916** |
| LOST [10] + Bilateral | .818 | .489 | .578 | .887 | .572 | .697 | .916 | .723 | .837 |
| TokenCut [11] + Bilateral | .897 | .618 | .697 | .914 | .624 | .755 | .934 | .772 | .874 |
| **MOVE (Ours) + Bilateral** | **.931** | **.636** | **.734** | **.951** | **.687** | **.821** | **.953** | **.801** | **.916** |
| SelfMask on pseudo* [12] | .923 | .609 | .733 | .938 | .648 | .789 | .943 | .779 | .894 |
| SelfMask on pseudo* [12] + Bilateral | **.939** | **.677** | **.774** | .949 | .694 | .819 | .951 | .803 | .911 |
| **SelfMask on MOVE (Ours)** | .933 | .666 | .756 | **.954** | **.728** | **.829** | **.956** | **.835** | **.921** |
| **SelfMask on MOVE (Ours) + Bilateral** | .937 | .665 | .766 | .952 | .687 | .827 | .952 | .800 | .917 |

*We found that SelfMask's $\max F_\beta$ metric was computed with an optimal threshold for each image instead of the entire dataset as in other methods; we re-evaluated their model for a fair comparison

(MAE) [21] with an adversarial loss to increase the details of the reconstructed images. For the discriminator we use the Projected Discriminator [24] in its standard setting, but we only use *color* differentiable augmentation. For the training we use random resized crops of size 224 with a scale in range $(0.9, 1)$ and aspect ratio $(3/4, 4/3)$. We set the minimum mask area $\theta_{\min} = 0.05$, the minimum loss coefficient $\lambda_{\min} = 100$ and we linearly ramp up the binarization loss coefficient $\lambda_{\mathrm{bin}}$ from 0 to 12.5 over the first 2500 segmenter iterations. We use the shift range $\Delta = 1/8$. We train the segmenter by alternatively minimizing the discriminator loss and the segmenter losses. Both are trained with a learning rate of 0.0002 and an Adam [27] optimizer with betas $= (0, 0.99)$ for the discriminator and $(0.9, 0.95)$ for the segmenter. We implemented our experiments in PyTorch [28]. We train our model for 80 epochs with a batch size of 32 on a single NVIDIA GeForce 3090Ti GPU with 24GB of memory.

## 4 Experiments

### 4.1 Unsupervised saliency segmentation

**Datasets.** We train our main model using the train split of the DUTS dataset (DUTS-TR) [30], containing 10,553 images of scenes and objects of varying sizes and appearances. We emphasize that we only use the images without the corresponding ground truth. For comparison, we evaluate our approach on three saliency detection datasets: the test set of DUTS (5,019 images), DUT-OMRON [29] (5,168 images) and ECSSD [31] (1,000 images). We report three standard metrics: pixel mask accuracy (Acc), intersection over union (IoU), $\max F_\beta$, where $F_\beta = \frac{(1+\beta^2)\mathrm{Precision}\times\mathrm{Recall}}{\beta^2\mathrm{Precision}+\mathrm{Recall}}$ for $\beta = 0.3$; the $\max F_\beta$ is the score for the single optimal threshold on a whole dataset. Additionally, we report the IoU on the test split [38] of CUB-200-2011 (CUB-Birds) [39] dataset.

**Evaluation.** We train our segmenter in an adversarial manner as specified in sections 2 and 3 and evaluate it on the test datasets. We compare with other methods in Table 1. Note that without any type of post-processing of our predicted masks, we surpass all other methods by a significant margin. We also follow [11, 12] and further refine our masks with a bilateral solver [40].

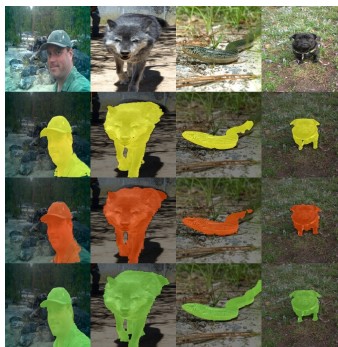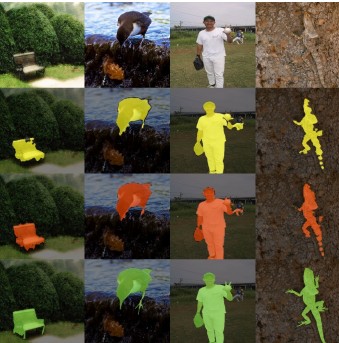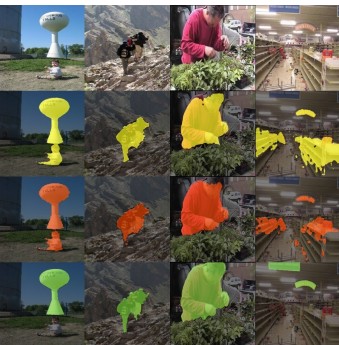

Figure 4: Qualitative evaluation of MOVE on ECSSD, DUTS-TE and DUT-OMRON. First row: input image; second row: MOVE; third row: SelfMask on MOVE; last row: ground truth. Best viewed in color. For more examples and a gray scale version see the supplementary material.

Since the bilateral solver only marginally improves or even decreases the quality of our segmentation, we conclude that our predicted masks are already very accurate. Using the bilateral solver might also inadvertently discard correct, but fragmented segmentations, as we show in the supplementary material. Next, we extract the predicted unsupervised masks from the DUTS-TR dataset and use them as pseudo ground-truth to train a class-agnostic segmenter. We use the same architecture (a MaskFormer [41]) and training scheme as SelfMask [12]. We then evaluate again on the saliency prediction datasets. Without additional pre-processing our method surpasses or is on par with the SotA across all metrics and datasets. While additional processing with the bilateral solver seems to benefit SelfMask [12], it mostly hurts the performance of our method. Figure 4 shows qualitative results of our method. Finally, we evaluate our method on the test set of CUB-Birds dataset. Additionally, we train our model on the train split of CUB-Birds dataset and run the same evaluation. We present the comparison with other methods in Table 2 and show that we achieve SotA performance.

Table 2: Comparison of unsupervised segmentation methods on the CUB-200-2011 test set. MOVE$^\star$ was trained on the CUB-200-2011 train set, while MOVE was trained on DUTS-TR

| Method | IoU |
|---|---|
| PerturbGAN [15] | 0.360 |
| ReDO [38] | 0.426 |
| IEM [18] | 0.551 |
| Melas-Kyriazi [37] | 0.664 |
| Voynov [36] | 0.683 |
| Voynov-E [36] | 0.710 |
| Deep Spectral [22] | 0.769 |
| **MOVE$^\star$** | **0.814** |
| **MOVE** | **0.858** |

## 4.2 Single-object discovery

**Datasets.** We evaluate our trained model (see section 4.1) on 3 typical single-object discovery benchmarks: the train split of COCO20K [42, 43] and the trainval splits of VOC07 [44] and VOC12 [45]. Following [10, 11, 43, 46–50], we report the *Correct Localization* metric (CorLoc), *i.e.*, the percentage of images, where the IoU > 0.5 of a predicted single bounding box with at least one of the ground truth ones.

**Evaluation.** Since our method tends to produce a single segmentation mask for multiple objects in the scene, we separate the objects by detecting connected components via OpenCV [51]. We then convert the separate masks to bounding boxes and choose the biggest one as our prediction for the given image. In Table 3, we compare MOVE with other unsupervised methods and we show that just by using processed masks from our method we achieve SotA results on all three datasets, outperforming even methods that used their bounding boxes to train a Class Agnostic Detector (CAD). We show qualitative results for object detection in Figure 5. We also follow the practice of [10, 11] and use our predicted bounding boxes as pseudo-ground truth for training the CAD on each of the evaluation datasets. To train the detector, we use either the largest or all the bounding boxes (*Multi*) that we obtained from the connected components analysis and after filtering those that have an area smaller than 1% of the image. For the evaluation we take the bounding box with the highest prediction confidence, as done in [10, 11]. We use the exact same architecture and training scheme as our competitors for a fair comparison. Training with a single bounding box improves the

Table 3: Comparisons for unsupervised single object discovery. We compare MOVE to SotA object discovery methods on VOC07 [44], VOC12 [45] and COCO20K [42, 43] datasets. Models are evaluated with the CorLoc metric. +CAD indicates training a second stage class-agnostic detector with unsupervised "pseudo-boxes" labels. (↑ z) indicates an improvement of z over prior sota

| Method | VOC07 [44] | VOC12 [45] | COCO20K [42, 43] |
|---|---|---|---|
| Selective Search [10, 52] | 18.8 | 20.9 | 16.0 |
| EdgeBoxes [10, 53] | 31.1 | 31.6 | 28.8 |
| Kim et al. [10, 54] | 43.9 | 46.4 | 35.1 |
| Zhange et al. [10, 55] | 46.2 | 50.5 | 34.8 |
| DDT+ [10, 56] | 50.2 | 53.1 | 38.2 |
| rOSD [10, 43] | 54.5 | 55.3 | 48.5 |
| LOD [10, 50] | 53.6 | 55.1 | 48.5 |
| DINO-seg [4, 10] | 45.8 | 46.2 | 42.1 |
| FreeSOLO [8] | 56.1 | 56.7 | 52.8 |
| LOST [10] | 61.9 | 64.0 | 50.7 |
| Deep Spectral [22] | 62.7 | 66.4 | 52.2 |
| TokenCut [11] | 68.8 | 72.1 | 58.8 |
| **MOVE (Ours)** | **76.0 (↑ 7.2 )** | **78.8 (↑ 6.7 )** | **66.6 (↑ 7.8 )** |
| LOD + CAD[10] | 56.3 | 61.6 | 52.7 |
| rOSD + CAD [10] | 58.3 | 62.3 | 53.0 |
| LOST + CAD [10] | 65.7 | 70.4 | 57.5 |
| TokenCut + CAD [11] | 71.4 | 75.3 | 62.6 |
| **MOVE (Ours) + CAD** | 77.1 | 80.3 | 69.1 |
| **MOVE (Ours) Multi + CAD** | **77.5 (↑ 6.1 )** | **81.5 (↑ 6.2 )** | **71.9 (↑ 9.3 )** |

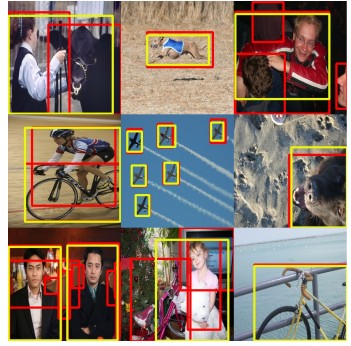 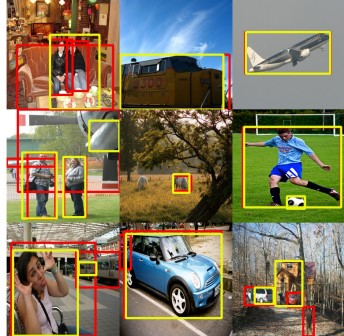 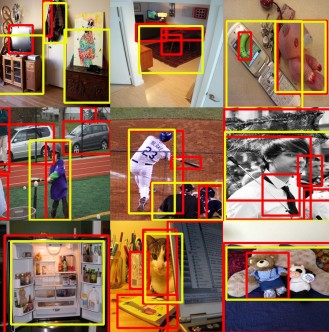

Figure 5: Qualitative evaluation of object detection of MOVE on VOC07, VOC12 and COCO20k. Red is the ground truth, yellow is our prediction. For more examples see the supplementary material.

performance of our method, while training with multiple ones gives it a significant additional boost.

**Unsupervised class-agnostic object detection.** We evaluate our unsupervised object detection model trained on COCO20K with CAD post-training and compare it with SotA on unsupervised class-agnostic object detection. In Table 4, we evaluate MOVE on COCOval2017 and report Average Precision (AP) and Average Recall (AR), as in [8]. MOVE yields a remarkable relative improvement over the AP SotA of 50% on average.

Table 4: Unsupervised class-agnostic object detection on MS COCO `val2017`. Compared results are taken directly from FreeSOLO [8]

| Method | $AP_{50}$ | $AP_{75}$ | AP | $AR_1$ | $AR_{10}$ | $AR_{100}$ |
|---|---|---|---|---|---|---|
| Sel. Search [52] | 0.5 | 0.1 | 0.2 | 0.2 | 1.5 | 10.9 |
| DETReg [57] | 3.1 | 0.6 | 1.0 | 0.6 | 3.6 | 12.7 |
| FreeSOLO [8] | 12.2 | 4.2 | 5.5 | 4.6 | 11.4 | 15.3 |
| **MOVE (Ours)** | **19.0** | **6.5** | **8.2** | **5.7** | **13.6** | **15.9** |

### 4.3 Ablation study

We perform ablation experiments on the validation split (500 images) of HKU-IS [58] to validate the relative importance of the components of our segmentation approach. For the ablation we train each

model for 80 epochs on DUTS-TR. We report the IoU in Table 5. Our baseline model trained with 3 different seeds gives a mean IoU 0.818 with std = 0.008. Thus we only report results for a single run in all experiments.

**Mask losses.** We validate the importance of the mask losses: minimum mask area, binarization and losses on downsampled max-pooled and avg-pooled masks. We find that the minimum area loss is necessary for our method to work, otherwise there is no incentive to produce anything other than empty masks. Removing the binarization loss or mask losses on the downsampled masks makes the masks noisier, which negatively affects the results.

**Shift range.** We evaluate different ranges of the random shift $\delta$. A small range $\Delta = 1/16$ makes it more challenging for the discriminator to detect inconsistencies at the border of objects. Larger shifts may cause objects to go out of the image boundaries ($\Delta = 3/16, 4/16$) and thus reduce the feedback at the object boundary to the segmenter. For $\Delta = 0$ (no-shift) the only possible discriminator inputs are composed images without a shift as fake and autoencoded images as real. There is no incentive to produce any meaningful masks in this case.

Table 5: Ablation study. Models evaluated on HKU-IS-val

| Setting | IoU |
|---|---|
| **Baseline (shift 2/16)** | **0.819** |
| no min. mask | 0.000 |
| no binarization loss | 0.774 |
| no pooled mask losses | 0.811 |
| no shift | 0.000 |
| shift 1/16 | 0.751 |
| shift 3/16 | 0.799 |
| shift 4/16 | 0.704 |
| disc. fake inputs: composed | 0.789 |
| disc. real inputs: $x$ + comp. w/o shift | 0.740 |
| disc. real inputs: comp. w/o shift | 0.031 |
| disc. real inputs: $x_{ae}$ | 0.000 |
| non-diff inpainter | 0.314 |
| MSE MAE | 0.817 |
| MAE feature extractor | 0.783 |
| ImageNet100 dataset | 0.815 |

**Discriminator inputs.** In our baseline model, we feed both composed images with no-shift and real images autoencoded with MAE as real samples and composed images with a shift and autoencoded images with copy-pasting of a predicted masked object as fake samples for the discriminator training. We test the case DISC. REAL $x$ + COMP. W/O SHIFT , where we feed to the discriminator real images without autoencoding. In this case, the discriminator can detect the artifacts of MAE instead of focusing on inconsistencies resulting from an incorrect mask. In DISC. REAL $x_{ae}$ we only feed the autoencoded images as real. Here, the discriminator can focus on the mismatch from the inpainting artifacts and encourages the segmenter to output empty masks, where no inpainting is done. If we only feed the composite non-shifted images (DISC. REAL COMP W/O SHIFT), the artifacts resulting from an incorrect masks cannot be fixed, because there is no reference of what the real images look like. In DISC. FAKE INPUTS: COMPOSED we only feed the composed image as fake to the discriminator and omit the real image with a copy-pasted predicted masked object, which slightly degrades the performance.

**Non-differentiable inpainter.** We evaluate the use of hard thresholded downsampled masks as input to the background inpainter. In this case the only feedback for the masks comes from the composition of the images. We find it to be insufficient for the segmenter to learn any meaningful masks.

**Inpainter model.** We substitute the MAE trained with a GAN loss with a MAE that was trained only to reconstruct missing patches with a Mean Squared Error (MSE) loss. Since this model was trained to only reconstruct the missing patches and not the entire image, we construct the inpainted background by composing the inpainted part with the real image: $\hat{m}_{up} = \text{upsample}_{16}(\hat{m})$; $\hat{b} := x \odot (1 - \hat{m}_{up}) + \hat{b} \odot \hat{m}_{up}$. Consequently, we do not use autoencoding when creating the discriminator inputs. We find this model to perform competitively.

**Feature extractor.** We train the model using the features provided by MAE encoder instead of a separate DINO model. In this case we adapted the segmenter architecture and added one more upsampling block, since MAE takes patches of size $P = 16$ (instead DINO has $P = 8$). We find that with these features we are able to train a competitive segmenter.

**ImageNet100 dataset.** We train our model on the ImageNet100 dataset [59], with 131,689 images from 100 randomly selected ImageNet [25] classes. Since this dataset is much bigger than DUTS-TR, we adapt our segmenter by adding an additional convolutional layer in each upsampling block (see section 3) and train the model for 8 epochs. The results are comparable to the DUTS-TR dataset.

# 5   Prior Work

In the past decade, research on object segmentation and detection has seen remarkable progress when full supervision is available [60–62]. To limit the cost of annotation several methods explored different forms of weak supervision [63, 64] or ways to avoid labeling altogether [13, 14, 65, 66]. MOVE falls in the latter category. Therefore, we focus our review of prior work on unsupervised methods for object segmentation and the related task of object detection.

**Unsupervised Object Detection and Category Discovery.** Unsupervised object detection and category discovery are extremely challenging tasks that have recently seen a surge of efforts [57, 67, 68] thanks to the capabilities of modern deep learning models. Recently, features based on deep learning have shown significant progress in object detection [57], even with just some noisy (unsupervised) guidance [52]. More in general, one limitation of unsupervised object detection is that it only provides a coarse localization of the objects. As we have shown with MOVE, it is possible to obtain much more information without supervision.

**Unsupervised Object Segmentation.** Object segmentation can be formulated as a pixel-wise image partitioning task [65, 66, 69, 70] or through the generation of layered models from which a segmenter is trained as a byproduct [8, 35, 71, 72]. The use of SSL features has spawned several methods with significant performance on real datasets, which we discuss in the following paragraphs.

**SSL-Based Methods.** Due to the success of SSL methods and the emergence of segmentation capabilities, several recent methods for unsupervised object segmentation have been built on top of SSL features. In particular, SelfMask [12] proposes a clustering approach that can use multiple SSL features and evaluates all possible combinations of DINO [4], SwAV [73] and MOCOV2 [74]. They find that combining features from all three SSL methods yields the best results for segmentation. FreeSOLO [8] instead finds that DenseCL features [6] work best. More in general, some methods use a weak (unsupervised) guidance and losses robust to the coarse pseudo-labels [8], but the majority is based on directly clustering SSL features [7, 9–11, 22, 75]. In contrast to these methods, we show that movability can provide a robust supervision signal.

**Generative Methods.** A wide range of methods also exploits generative models to create layered image representations [15, 17, 20, 76–79]. A general scheme is to train a network to generate a background, a foreground and its mask. These components can then be combined to generate an image and then, in a second stage, one can train a segmenter that learns to map a synthetic image to its corresponding foreground mask. In alternative, the segmenter could be built during the training of the generative model as a byproduct. Some methods rely on the assumption that a dataset of only backgrounds is available [14, 80]. The use of shifts to define segments has also been used before [13, 15, 16]. However, MOVE does not require the training of a generative model, which can be a challenge on its own.

# 6   Limitations and Societal Impact

**Limitations.** As mentioned in the introduction, movability alone may not suffice in identifying an object unambiguously. In fact, the method can segment any combination of multiple objects. To address this we use a post-processing algorithm to find connected components, but there is no guarantee that all objects have been segmented. Another issue is that shifts would not expose artifacts when the background is uniform (*e.g.*, looking at the sky, underwater, with macrophotography).

**Societal Impact.** An important aspect that is relevant to unsupervised learning methods in general is the potential to become biased if the training datasets are unbalanced. This clearly has a potentially negative impact on the segmentation of categories that are underrepresented and thus this work should be integrated with mechanisms to take the dataset imbalance into account.

# 7   Conclusions

We have introduced MOVE, a novel self-supervised method for object segmentation that exploits the synthesis of images where objects are randomly shifted. MOVE improves the SotA in object saliency segmentation, unsupervised single object discovery, and unsupervised class agnostic object detection by significant margins. Our ablations show that movability is a strong supervision signal that can be robustly exploited as a pseudo-task for self-supervised object segmentation. We believe that our approach can be further scaled by exploring different architectures and larger datasets.

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
