# Supplementary material

**Adam Bielski**
University of Bern
adam.bielski@unibe.ch

**Paolo Favaro**
University of Bern
paolo.favaro@unibe.ch

We report further evaluations of MOVE that could not be included in the main paper. An important component of MOVE is the Masked Autoencoder. We perform additional analysis to show how MAE encodes image tiles and also how this changes between a GAN or MSE trained MAE. We also examine the mask losses when they are used at multiple scales. In other methods, the bilateral solver is used to refine the predicted masks. We show that our segmenter produces already accurate masks even without the bilateral filter. Finally, we show additional quantitative and qualitative results on other training and test datasets.

Table A.1: Inpainting error for a pre-trained MAE on 5000 images from the ImageNet validation set: Feeding a subset of tokens to the encoder (Default) vs soft-masking before the decoder (Modified). $\Delta$ is the mean squared error between the inpainted regions for two methods

| MAE Model | Default | Modified | $\Delta$ |
|---|---|---|---|
| w/ GAN | $0.0683 \pm 0.0427$ | $0.0647 \pm 0.0398$ | $0.0070 \pm 0.0059$ |
| w/ MSE | $0.0639 \pm 0.0411$ | $0.0617 \pm 0.0390$ | $0.0055 \pm 0.0056$ |

## A  MAE as a differentiable inpainter

Masked Autoencoders (MAE) consist of a Transformer Encoder, which takes as input only a subset of unmasked patches during training, and a Transformer Decoder, which takes as input the encoded patches and, in addition, a learnable MSK token replicated at all the locations where the (masked) patches were not fed to the encoder. The decoder is trained to reconstruct the masked patches.

In MOVE, we need the pre-trained MAE to work as a differentiable inpainter. To that end, we feed all the patches to the encoder. Then, we only do a soft-masking between the MSK token and the encoded patches via a convex combination, before feeding the embeddings to the decoder (see section 2 and Figure 3). This is different from how MAE was trained: During training the encoder had no way to encode the information about the missing patches. Since in MOVE we feed all the patches to the encoder, it is possible that the encoded embeddings contain information about their neighbors. In particular, there is the risk that the unmasked encoded patches would contain information about the masked patches. If that were the case, the decoder would be able to inpaint the masked object even when the entire object is masked at the decoder input. We show empirically and quantitatively that this is not the case. Using the same pre-trained MAE, we compare the reconstruction error for the original inference vs. our modified soft-masking inference. We run the evaluation on a subset of 5000 images from the ImageNet validation set [1], randomly masking between $80\%$ and $95\%$ of the tokens. We show the mean squared error of the intensity for intensity range $[0; 1]$ in Table A.1 and comparison of reconstructed images in Figure A.1 for both MAE trained with a GAN loss or with an MSE loss. We find that the difference in the inpainting error is not significant. Moreover, we observe visually that the reconstructions through the Modified soft-masking (MOVE) do not show a better reconstruction of the masked patches than in the Default case where the masked patches are not provided to MAE.

36th Conference on Neural Information Processing Systems (NeurIPS 2022).

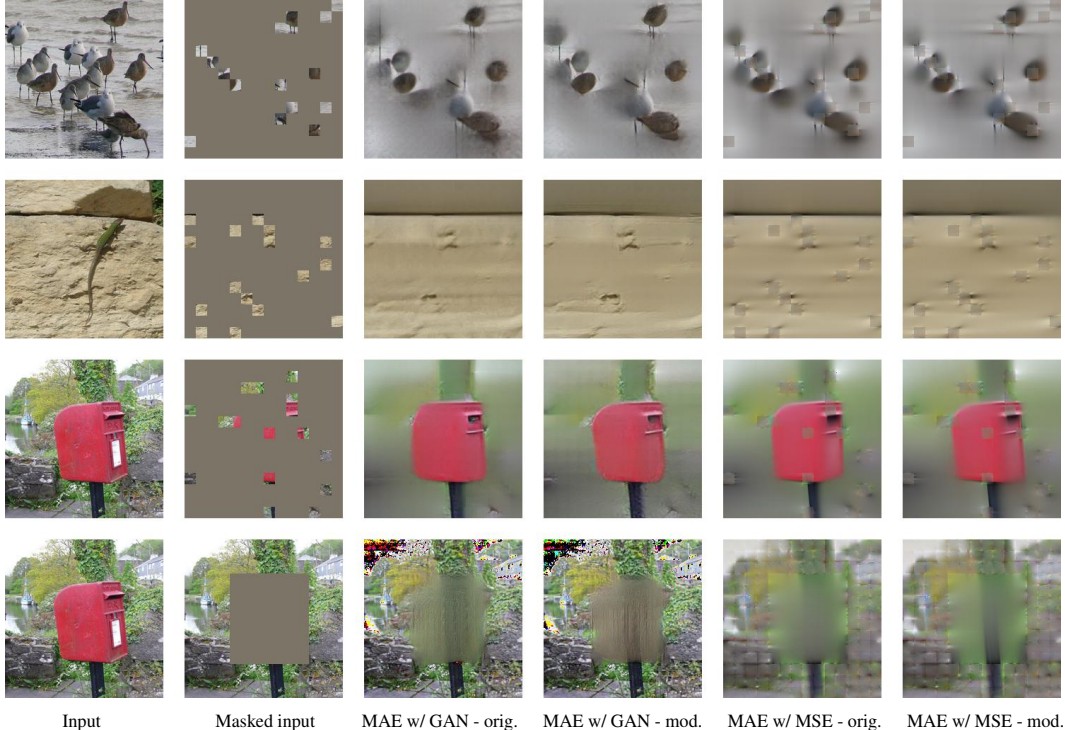

| Input | Masked input | MAE w/ GAN - orig. | MAE w/ GAN - mod. | MAE w/ MSE - orig. | MAE w/ MSE - mod. |

Figure A.1: Comparison of MAE sparse input vs differentiable mask inpainting. We show the input and masked input image in the two first columns. For MAE trained with a GAN loss or with an MSE loss we show the reconstructed image when we feed a sparse subset of tokens to the encoder (orig.) and when we feed all the tokens to the encoder and mask only before feeding the embeddings to the decoder (mod.). No significant difference can be observed between these two reconstruction modalities or when we change the MAE training.

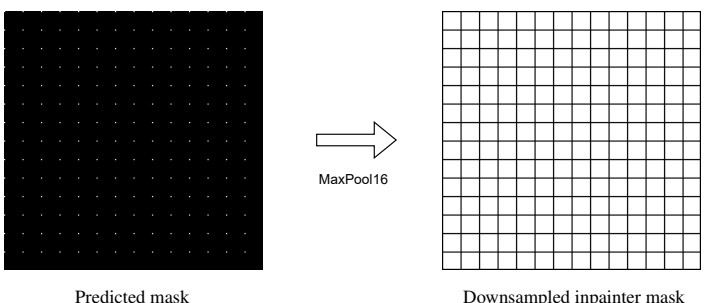

Predicted mask                    Downsampled inpainter mask

Figure B.1: Obtaining an inpainting mask from a predicted mask via max pooling downsampling. Due to small artifacts in the mask, all patches might be selected as masked and thus, the entire background might get inpainted. The grid on the right is just for reference purposes.

## B  Inpainter mask and downsampled mask losses

As specified in section 2, we obtain a low-res inpainting mask $\hat{m}$ via a $\text{maxpool}_P$ with stride $P$ operation on the union of the predicted mask and its shifted version, where $P$ is the patch size that MAE tokens embed. We use max pooling for downsampling, because we want to make sure that we mask all the patches containing even only parts of the object. This is important, otherwise the inpainter may partly reconstruct the object. However, using max pooling for downsampling might result in inpainting more than necessary due to the artifacts in the mask. An extreme case of this is illustrated in Figure B.1, where the entire background would get inpainted due to a single pixel

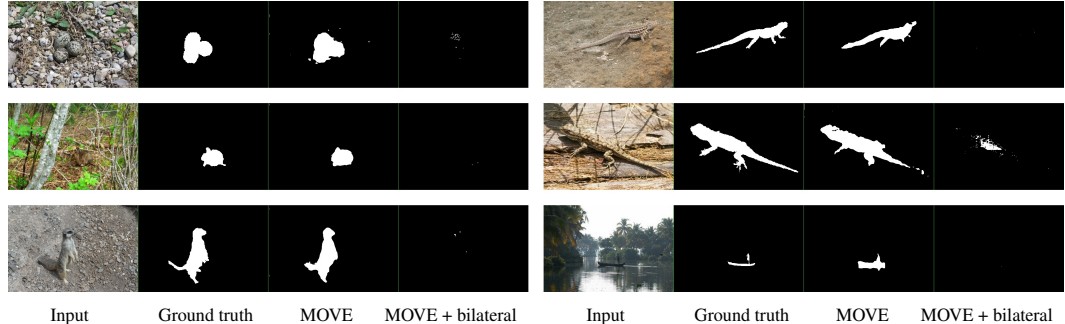

| Input | Ground truth | MOVE | MOVE + bilateral | Input | Ground truth | MOVE | MOVE + bilateral |

Figure C.1: A refinement with the bilateral solver might cause the shrinking of valid predicted masks.

within each $P \times P$ patch. To avoid such cases we apply our $\mathcal{L}_{\min}$ and $\mathcal{L}_{\text{bin}}$ losses (eq. (1),(2)) on the downsampled mask as well. Having a binarization loss on the mask downsampled with max pooling has an extra regularizing effect on the original mask. For example, when all mask pixels in a patch have a value below 0.5, the binarization loss on the max pooling of the mask will push only the largest value towards 0. This creates an asymmetry when the pixels of the mask must be reduced, which prioritizes the largest values. Eventually however, the application of this loss over multiple iterations will result in pushing all pixels within the patch to 0.

## C   Bilateral solver

While other methods get competitive results by using a bilateral solver to refine the masks predicted from their methods (see section 4.1), MOVE provides more accurate results wihout any additional post-processing. The application of a bilateral solver, which highly relies on image texture, could even decrease the performance in cluttered images. In Figure C.1 we show some examples where the bilateral solver hurts our predictions.

## D   Segmenter architecture

Our segmenter is built on top of a ViT-based feature extractor, as specified in section 3. We define a $\text{Block}_{out\_ch}^{in\_ch}$ as a sequence of layers:

$$3 \times 3 \, \text{Conv}_{out\_ch}^{in\_ch} \to \text{BatchNorm} \to \text{LeakyReLU}, \tag{1}$$

where $K \times K \, \text{Conv}_{out\_ch}^{in\_ch}$ is a padded $K \times K$ convolution with stride $= 1$, $in\_ch$ input channels and $out\_ch$ output channels. Our baseline segmenter takes DINO ViT-S/8 384-dimensional features arranged in a grid and consists of alternating upsampling layers and blocks:

$$\begin{aligned}
\text{Upsample}_{\text{nearest}}^{2\times2} &\to \text{Block}_{192}^{384} \to \\
\text{Upsample}_{\text{nearest}}^{2\times2} &\to \text{Block}_{128}^{192} \to \\
\text{Upsample}_{\text{nearest}}^{2\times2} &\to \text{Block}_{128}^{128} \to \\
\text{Block}_{128}^{128} &\to 1 \times 1 \, \text{Conv}_1^{128}
\end{aligned} \tag{2}$$

MAE features used as the segmenter inputs in one of the ablations (section 4.3) are 1024-dimensional come from ViT/16 with $16 \times 16$ patches, therefore the segmenter needs an extra upsampling block and adapted number of channels. The adapted architecture in this case is

$$\begin{aligned}
\text{Upsample}_{\text{nearest}}^{2\times2} &\to \text{Block}_{512}^{1024} \to \\
\text{Upsample}_{\text{nearest}}^{2\times2} &\to \text{Block}_{256}^{512} \to \\
\text{Upsample}_{\text{nearest}}^{2\times2} &\to \text{Block}_{128}^{256} \to \\
\text{Upsample}_{\text{nearest}}^{2\times2} &\to \text{Block}_{128}^{128} \to \\
\text{Block}_{128}^{128} &\to 1 \times 1 \, \text{Conv}_1^{128}
\end{aligned} \tag{3}$$

For the ImageNet100 experiment we increase the capacity of the segmenter by making each Block deeper, i.e. $\text{Block}_{out\_ch}^{in\_ch}$ is:

$$3 \times 3 \text{ Conv}_{out\_ch}^{in\_ch} \rightarrow \text{BatchNorm} \rightarrow \text{LeakyReLU} \rightarrow$$
$$3 \times 3 \text{ Conv}_{out\_ch}^{out\_ch} \rightarrow \text{BatchNorm} \rightarrow \text{LeakyReLU}. \tag{4}$$

# E   Additional results

## E.1   Segmentation qualitative results

In Figures E.1,E.2,E.3 we show more segmentation results of MOVE on DUTS-TE, DUT-OMRON and ECSSD.

## E.2   Detection qualitative results

In Figures E.4,E.5,E.6 we show more object detection results of MOVE on VOC07, VOC12 and COCO20k.

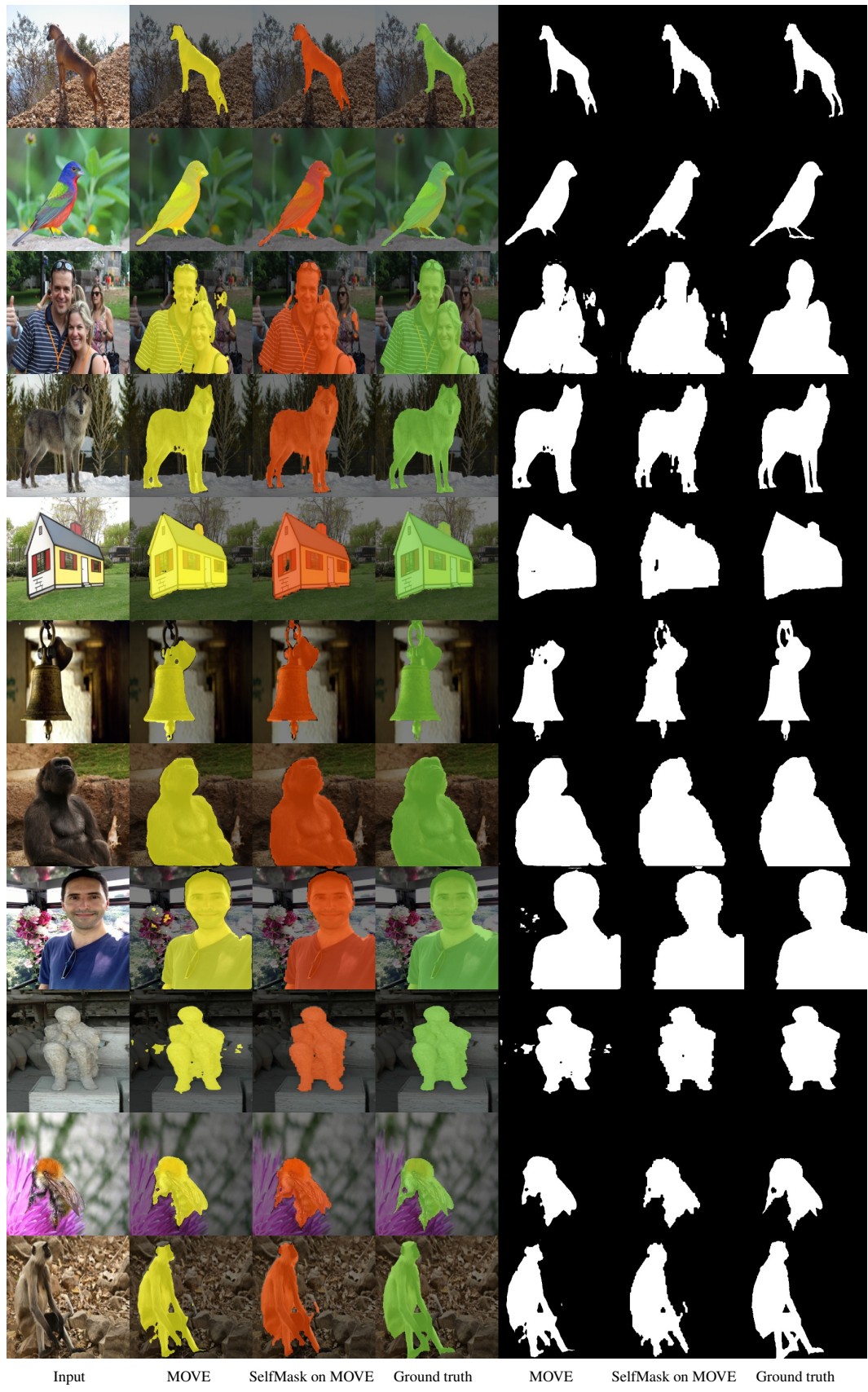

Input     MOVE     SelfMask on MOVE     Ground truth       MOVE     SelfMask on MOVE     Ground truth

Figure E.1: Sample segmentation results on ECSSD.

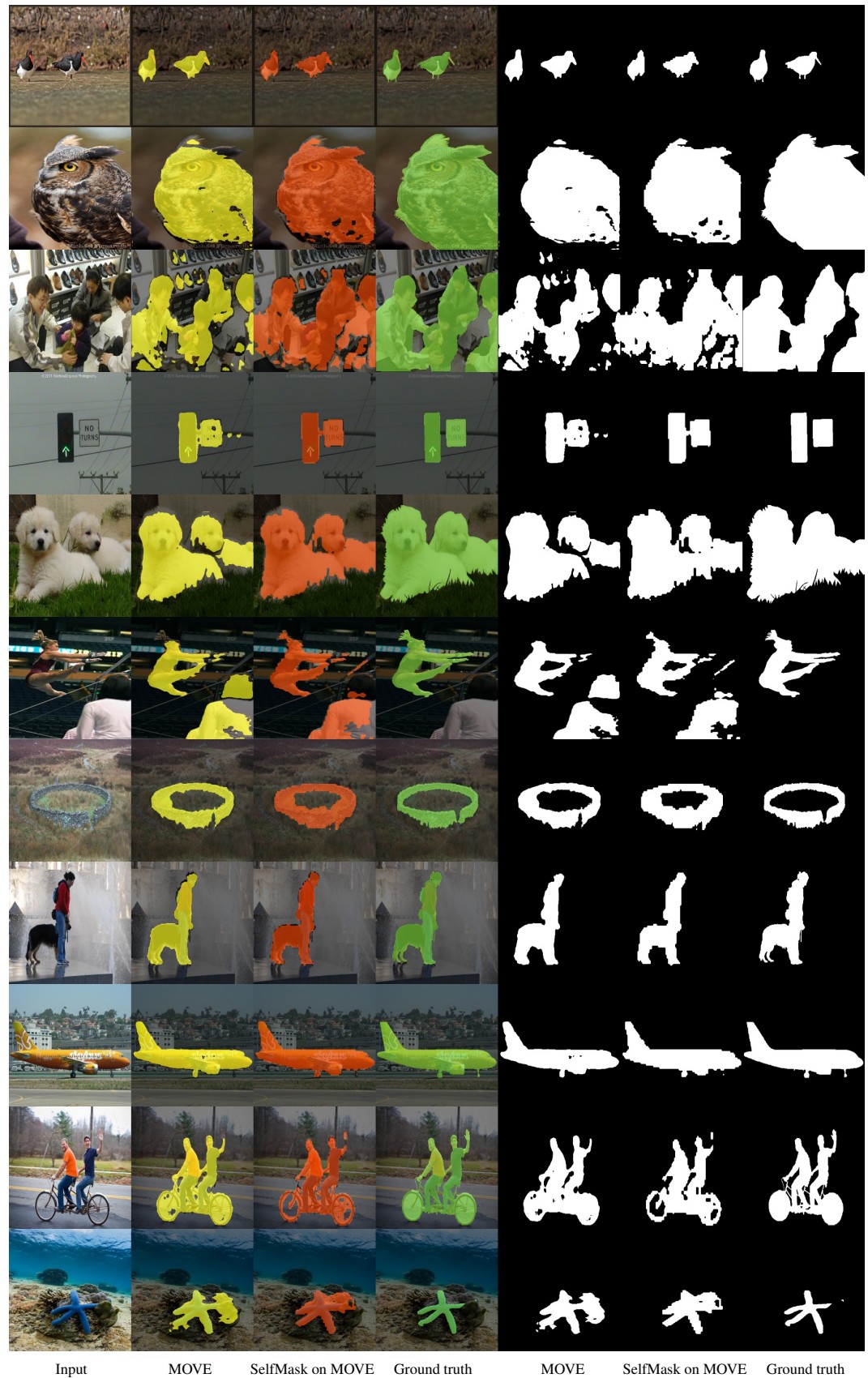

Input      MOVE      SelfMask on MOVE    Ground truth      MOVE      SelfMask on MOVE    Ground truth

Figure E.2: Sample segmentation results on DUTS-TE.

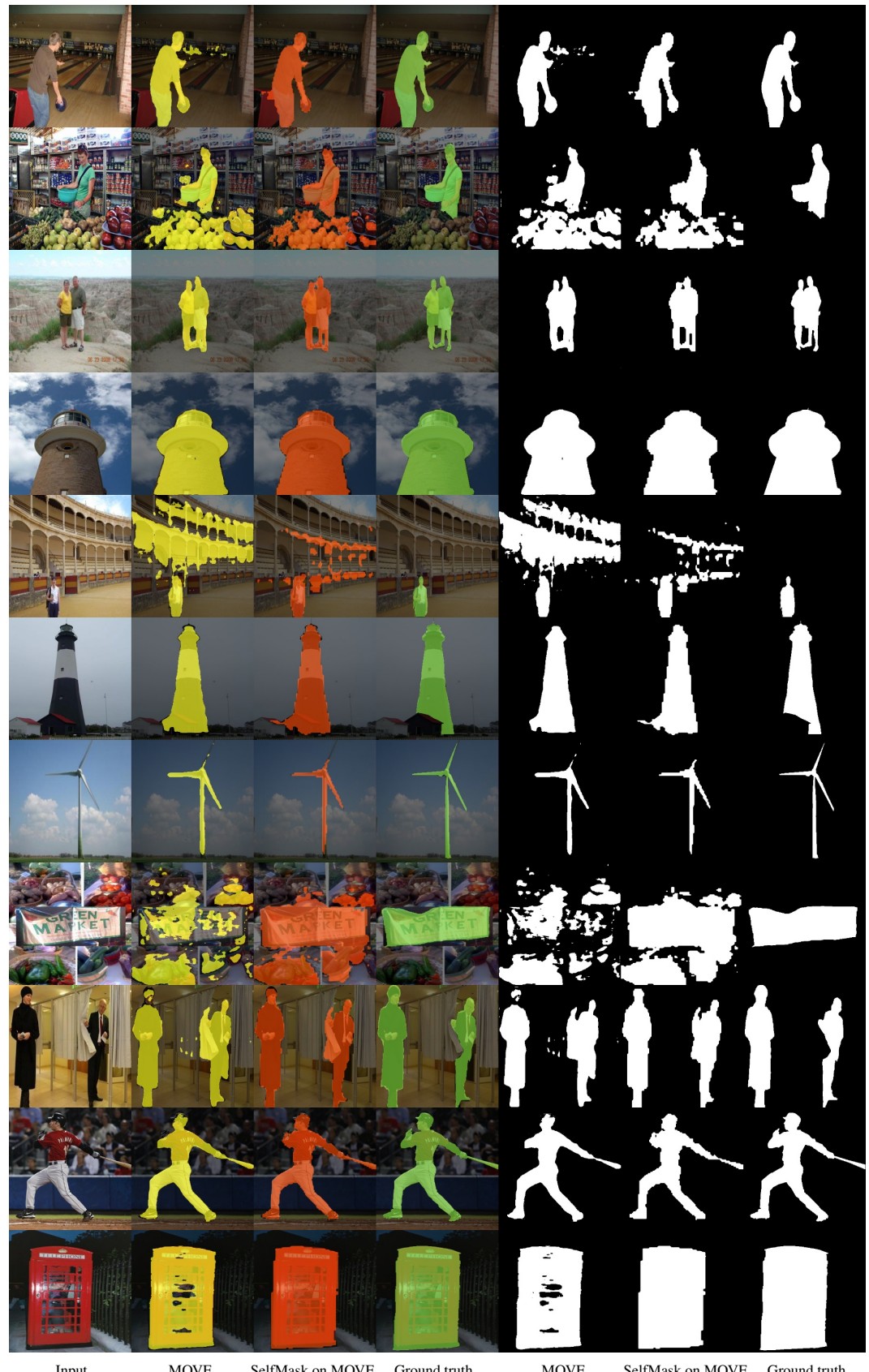

Input        MOVE     SelfMask on MOVE   Ground truth       MOVE     SelfMask on MOVE   Ground truth

Figure E.3: Sample segmentation results on DUT-OMRON.

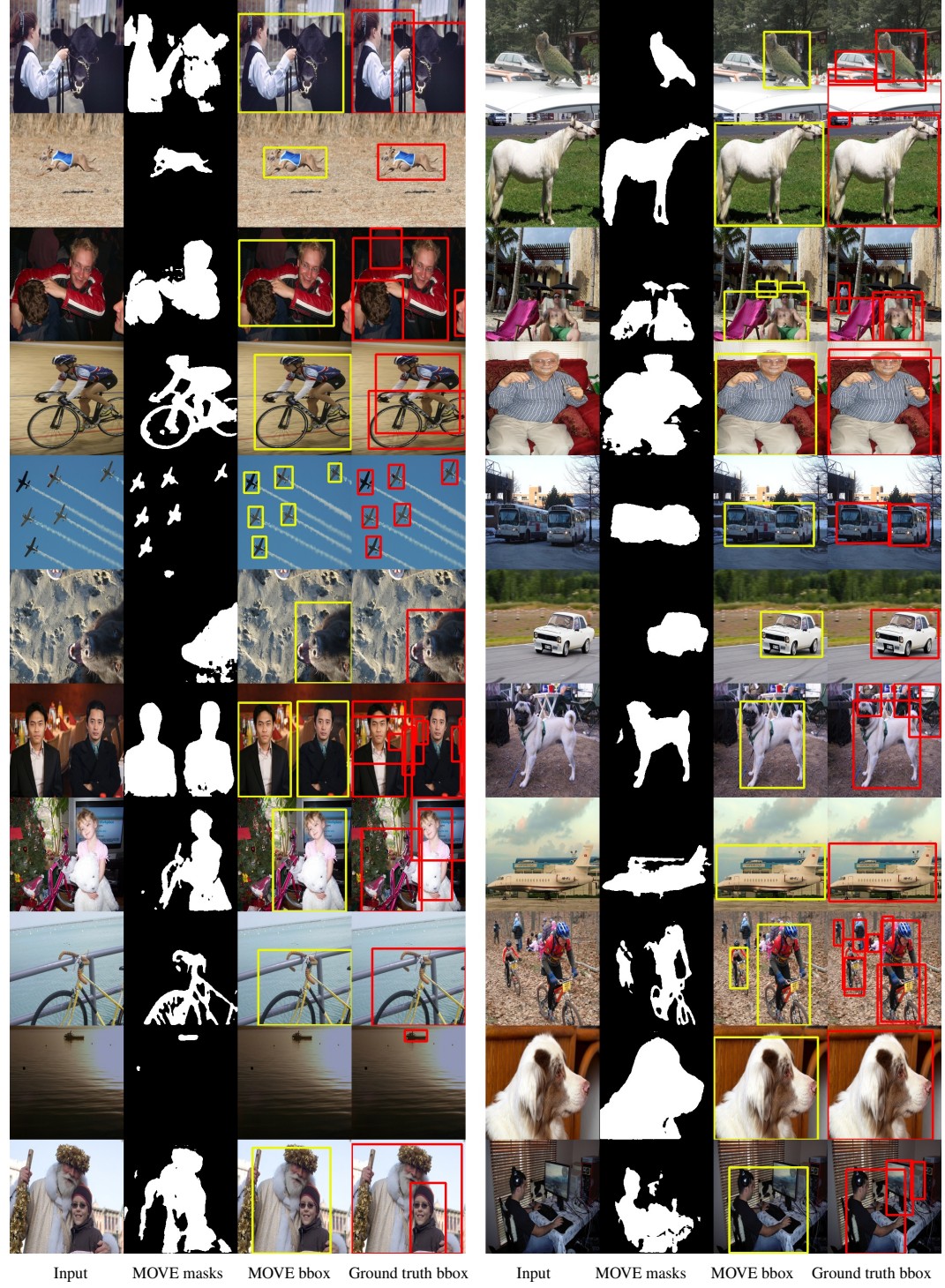

Input  MOVE masks  MOVE bbox  Ground truth bbox  Input  MOVE masks  MOVE bbox  Ground truth bbox

Figure E.4: Sample detection results on VOC07.

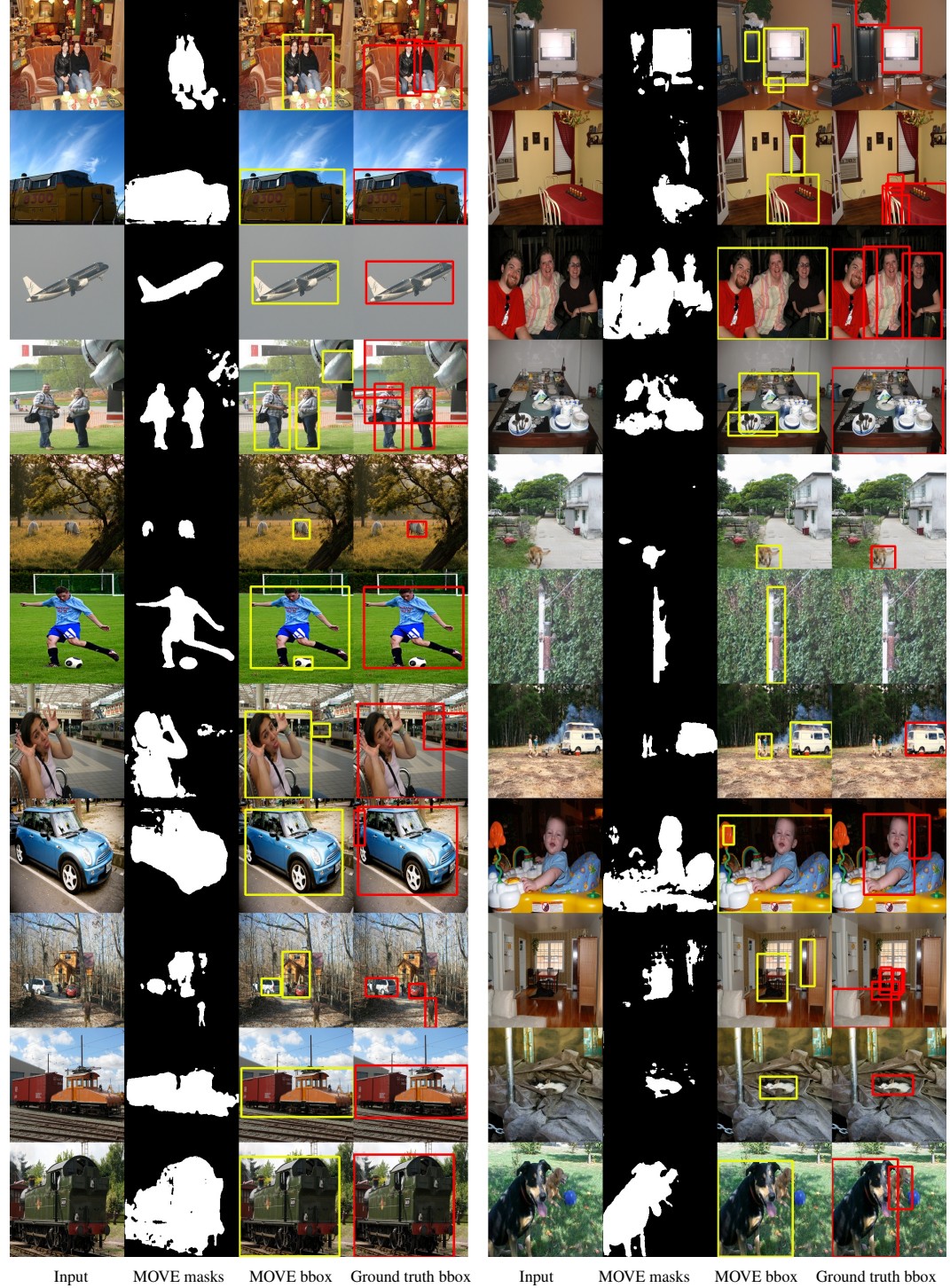

Input     MOVE masks     MOVE bbox     Ground truth bbox        Input     MOVE masks     MOVE bbox     Ground truth bbox

Figure E.5: Sample detection results on VOC12.

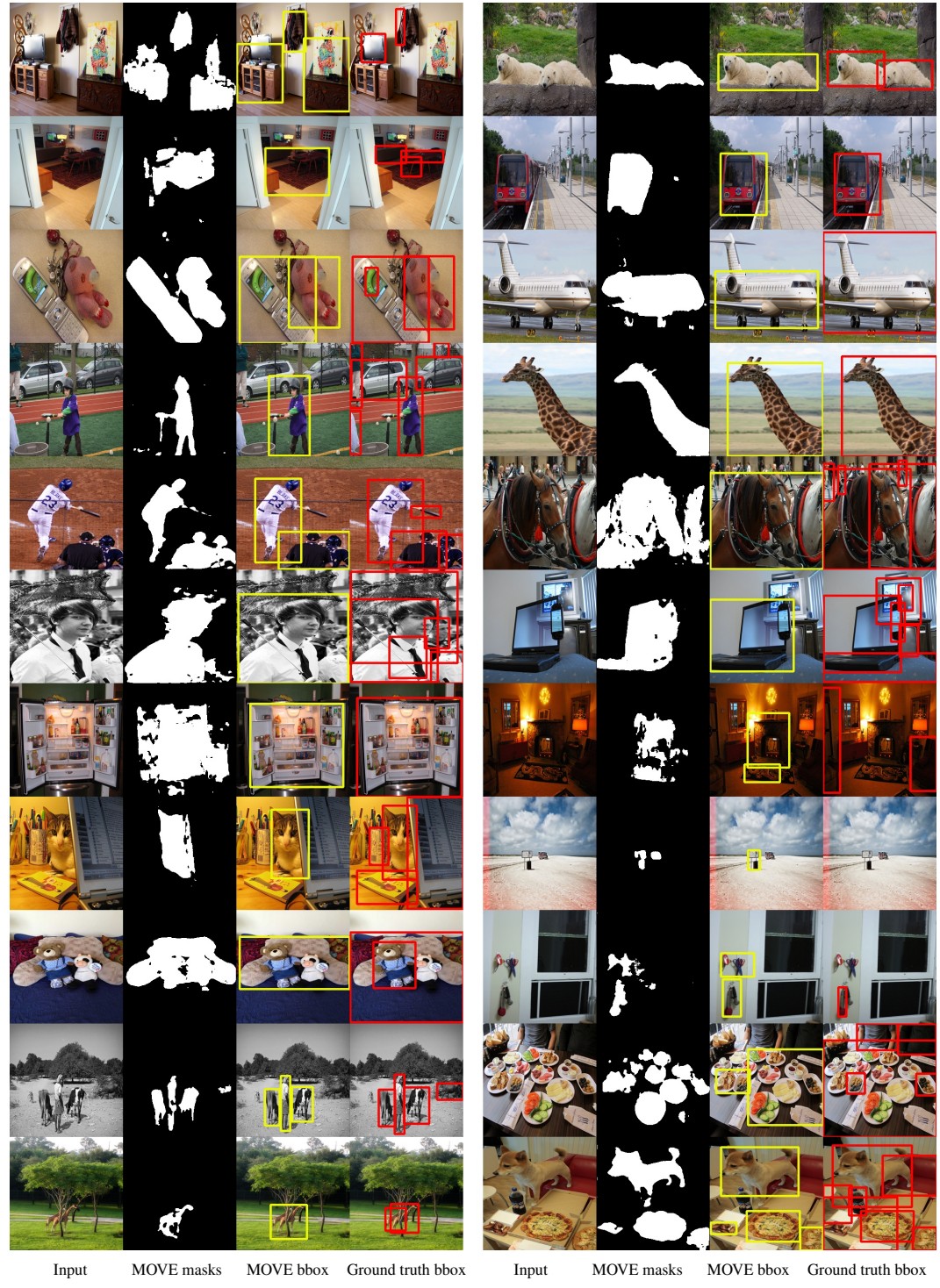

Input     MOVE masks     MOVE bbox     Ground truth bbox          Input     MOVE masks     MOVE bbox     Ground truth bbox

Figure E.6: Sample detection results on COCO20k.

# References

[1] Jia Deng, Wei Dong, Richard Socher, Li-Jia Li, Kai Li, and Li Fei-Fei. Imagenet: A large-scale hierarchical image database. In *2009 IEEE Conference on Computer Vision and Pattern Recognition*, pages 248–255, 2009. doi: 10.1109/CVPR.2009.5206848.