# OpenReview forum: "MOVE: Unsupervised Movable Object Segmentation and Detection"
_NeurIPS.cc/2022/Conference — NeurIPS 2022 Accept_

### Official Review · Reviewer_tSAq · 2022-07-06

**Rating:** 8
**Confidence:** 4
**Soundness:** 4 excellent
**Presentation:** 4 excellent
**Contribution:** 4 excellent

**Summary:**

This work introduces a framework for unsupervised image segmentation. The assumption that it exploits is that a slightly shifted foreground segment overlayed on an inpainted background image, whenever the former is meaningful, must look realistic. The work develops and carefully implements this idea using a transformer to generate and inpaint the masks. A discriminator network is tasked to distinguish between the generated images and the real images, where appropriate choices were made to avoid degenerate solutions by augmenting the real image distribution. Elaborate experiments on saliency detection, single object discovery and detection, demonstrate substantially improved accuracy compared to previous state of the art.

Overall, the paper is insightful, original and technically sound. It makes a clear advance in the problem of unsupervised segmentation.

**Post-rebuttal comment.**
I thank the authors for futher elaborations. Regarding the concern of *R-tM6J*: I agree that using pre-trained features (or even depending on it) is a limitation. However, I do not see it as critical. After all, the pre-training is completely unsupervised, and performing accurate segmentation on the representation learned – especially to the degree demonstrated in this work – is highly non-trivial.

In short, this is a nice work that deserves to be presented at the conference, which I strongly recommend.

*To the authors:*
- Please do elaborate on the methodology for tuning the framework, reported in Tab. 1&2, in the revision. This increasingly  becomes a source of confusion in the literature on unsupervised learning, impedes reproducibility and fairness of comparisons.
- The ablation Table 4 needs some revision. Adding some clarification in the caption, or changing the table layout may help.

**Questions:**

A few open questions:
- Since the framework is completely unsupervised, what stands in the way of training it on larger datasets (e.g. ImageNet)? Would it scale?
- What criteria and data were used to choose hyperparameters?
- How crucial is the ImageNet self-supervised pre-training of the feature encoder?

**Limitations:**

Limitations and the societal impact are discussed at appropriate length.

**Strengths And Weaknesses:**

Strengths:
* clearly written;
* contains a number of technically innovative bits, since implementing this conceptually simple idea is actually prone to degenerate solutions;
* extensive experimentation with strong results across the board.

A minor weakness:

The main idea exploited here re-implements the “cut-and-paste” principle from previous studies, but in the context of the latest advances in network architectures (ViT), self-supervised learning (DINO) and autoencoders (MAE). However, this is given appropriate credit in the text (l. 29). Also, the focus on small shifts, in contrast to some previous works, also makes sense geometrically, since perspective distortions can be neglected.


A few suggestions:

Fig. 2 and 3 are disconnected and show different parts of the pipeline. Having a unified “overview” diagram, that also depicts
- what images the “fake” and “real” sets comprise (as the input to the discriminator),
- which network modules remain fixed and which are trained,
would help in accompanying the text in Sec. 2, which is is quite dense.

In Tab. 1-3, it would interesting to see an overview of the computational footprint of those methods (e.g. the backbone / FLOPS) and/or the datasets used for training (e.g. “ImageNet + DUTS-TR”) for this work.

Images in Fig. 4/5 are too small. I’d rather use fewer samples, but in larger size. More images can be always added to the supplemental.

---

> ### Author Response · Authors · 2022-08-02
> **Response to Reviewer tSAq**
>
> We thank the reviewer for the valuable feedback and useful comments.
>
> Below we'd like to answer the questions raised by the reviewer.
>
> -   *Fig. 2 and 3 are disconnected and show different parts of the pipeline. Having a unified "overview" diagram, that also depicts:*
>
>     *- what images the "fake" and "real" sets comprise (as the input to the discriminator),*
>
>     *- which network modules remain fixed and which are trained, would help in accompanying the text in Sec. 2, which is is quite dense.*
>
>     -   This is a very good suggestion. We will integrate an illustration as described above in the paper.
>
>
> -   *In Tab. 1-3, it would interesting to see an overview of the computational footprint of those methods (e.g. the backbone / FLOPS) and/or the datasets used for training (e.g. "ImageNet + DUTS-TR") for this work.*
>
>     -   We were able to train MOVE on bigger datasets and we plan to include these results in the next revision of the paper.
>
> -   *Images in Fig. 4/5 are too small. I'd rather use fewer samples, but in larger size. More images can be always added to the supplemental.*
>
>     -   We agree. We'll show fewer and larger images.
>
> -   *Since the framework is completely unsupervised, what stands in the way of training it on larger datasets (e.g. ImageNet)? Would it scale?*
>
>     -   After the submission we were able to train MOVE on bigger datasets, including Pascal VOC, COCO20k, ImageNet100 and ImageNet, obtaining similar results to DUTS-TR used in the paper. We suspect that we'd need to carefully scale the architecture as well to be able to benefit from bigger datasets. This is indeed a direction for future work.
>
> -   *What criteria and data were used to choose hyperparameters?*
>
>     -   We ran some of the evaluation on the validation set of HKU-IS dataset (as described in the ablation study).
>
> -   *How crucial is the ImageNet self-supervised pre-training of the feature encoder?*
>
>     -   Using a pre-trained self-supervised feature encoder allowed us to train MOVE on small datasets and fast. We tried an experiment with a different feature encoder, i.e. using features from the MAE encoder (the same ones that are used for inpainting) and we were able to train the model with only a slight decrease in performance. It shows that we are not relying specifically on DINO features. We tried to train MOVE by allowing to fine-tune the entire DINO ViT-S-8 network, but the model failed to produce anything meaningful in this case. We believe that the entire segmentation model could be trained from scratch. However, it would be computationally demanding (eg, require a resource-heavy architecture, hyperparameter search and a longer training schedule).

---

### Official Review · Reviewer_tM6J · 2022-07-07

**Rating:** 3
**Confidence:** 5
**Soundness:** 2 fair
**Presentation:** 1 poor
**Contribution:** 1 poor

**Summary:**

The paper proposes an unsupervised segmentation approach using the "movability" as the cue to differentiate between foreground and background. The pretext task is to segment the foreground object, shift it into a new location, and compose the new image with inpainted backgrounds. The approach relies heavily on pretrained models, DINO for the segmentation network, and MAE for the inpainting network. A GAN loss is used to encourage the composed image to be realistic. The method assumes that the segmentation network has to segment the foreground object in order to synthesize realistic images. Encouraging results are reported on saliency detection and unsupervised object discovery.

**Questions:**

- I think it is inappropriate to use the term "movability" to describe such method. It would confuse readers that the method exploit motion cues to segment objects in videos. It is also not correct that any foreground object could be displaced to any location. For example, the monkey could not be displaced to the sky. Also, other background objects or stuffs could be displaced properly. For example, the tree log could be moved left or right.

- In figure 3, the encoded foreground looks identical to that of the input foreground. Why do we have to encode foreground through MAE?

- Also in figure 3, the inpainted background looks unreal. The grassland behind monkey now becomes homogeneous green region. Why is this a meaningful task? The discriminator would only needs to find these inpainting artifacts in order to discriminate between real and fake. No matter predicted mask is, the inpainted results will always have artifacts which may impede GAN learning.


**Ethics Review Area:**

["I don’t know"]

**Limitations:**

The authors adequately addressed the limitations and potential negative societal impact of their work.

**Strengths And Weaknesses:**

Strengths:
+ The idea of using inpainting to formulate a self-supervised learning pretext task is interesting.

+ The empirical result is encouraging. Notable improvements are obtained against prior works on all benchmarks.

Weaknesses:
- The method relies heavily on pretrained models, yet without discussing the significance of it. I suspect that the method would not work at all without pretrained DINO & MAE models. The general claim of the method should work even without any pretrained models. I am not convinced about the paper it does not work standalone.

- The writing is poor, especially in the approach section page 4. Description of the method is disorganized.

---

> ### Author Response · Authors · 2022-08-02
> **Response to Reviewer tM6J**
>
> We thank the reviewer for the valuable feedback and comments. We'd like to clarify some points below.
> - *The method relies heavily on pretrained models, yet without discussing the significance of it. I suspect that the method would not work at all without pretrained DINO & MAE models. The general claim of the method should work even without any pretrained models. I am not convinced about the paper it does not work standalone.*
> 	- We find this feedback very confusing. All other unsupervised segmentation learning methods of CVPR 2022 (and earlier) build on top of pre-trained SSL features such as DINO. Nonetheless, these works are very exciting developments for such an extremely difficult task.
> 	- MOVE does not depend on the exact choice of DINO and MAE. We also successfully trained MOVE with either a GAN or L2-trained MAE inpainter. Also, we were able to train MOVE using MAE encoder features instead of DINO features as input to the trainable segmenter. We obtained only a slightly decreased performance. Using a pre-trained model allows us to train MOVE on smaller datasets and in fewer iterations. We believe that the entire segmentation model could be trained from scratch. However, it would be computationally demanding (eg, require a resource-heavy architecture, hyperparameter search and a longer training schedule).
> - *The writing is poor, especially in the approach section page 4. Description of the method is disorganized.*
> 	- We would appreciate a suggestion on how we can make the description clearer.
> - *I think it is inappropriate to use the term "movability" to describe such method. It would confuse readers that the method exploit motion cues to segment objects in videos.*
> 	- By the term "movability" we mean "being able to be moved" in the real world, which is one of the properties of physical objects. We'll make it clearer, we will say that we only work on single images and predict what can we moved, not what is moving.
> - *It is also not correct that any foreground object could be displaced to any location. For example, the monkey could not be displaced to the sky.*
> 	- The observation is correct and was already pointed out in the Cut&Paste work of Remez et al 2018. Objects cannot be displaced to any arbitrary location as it would result in a scene that doesn't look realistic. That is exactly why we are relying on local shifts for which the context of the shifted object stays approximately the same, as noted in section 2, line 112. Statistically, most of the local shifts give realistic scenes, so it does not matter that a few cases do not hold (neural networks are known to be robust to some degree of label noise).
> - *Also, other background objects or stuffs could be displaced properly. For example, the tree log could be moved left or right.*
> 	- As we point out in the Limitations section of our paper, there is some ambiguity in the selection of objects in our method and there is no guarantee that all objects will get segmented. This is a direction for future research.
> - *In figure 3, the encoded foreground looks identical to that of the input foreground. Why do we have to encode foreground through MAE?*
> 	- We explain it in the description of Figure 3 and section 2 lines 124-127. This is one of the important details to make the adversarial training work. The MAE auto-encoding is used to avoid discriminator shortcuts. The auto-encoded foreground may look the same to a human observer. However, MAE, as any other neural network, outputs images with artifacts that would make it easy for the discriminator to distinguish them from real images. Since the inpainted background is produced by MAE, we also autoencode the foreground before the composition, so that every pixel is processed by MAE once. For the same reason instead of directly feeding images as real samples to the discriminator, we first auto-encode them. We discuss the importance of discriminator inputs in the ablation study, section 4.3.
> - *Also in figure 3, the inpainted background looks unreal. (...) The discriminator would only needs to find these inpainting artifacts in order to discriminate between real and fake. No matter predicted mask is, the inpainted results will always have artifacts which may impede GAN learning.*
> 	- Indeed, we address and explain this in section 2, lines 115-124. Without our training techniques, the inpainted background would provide an easy clue for the discriminator that the composite image is fake. We create a second set of "real" images by composing the impainted background with the foreground _without any shift_. These images have the same amount of inpainting as those with the shift. We feed them as another set of "real" images to the discriminator, so that the discriminator focuses on artifacts exposed by shifting inaccurately segmented objects, rather than artifacts from the inpainter. We show in the ablation study (section 4.3, lines 226-228) that the training fails if we only feed the autoencoded real images.

---

> > ### Comment · Reviewer_tM6J · 2022-08-03
> > **thanks for your responses**
> >
> > I thank the authors for providing a rebuttal. I have carefully read through your comments and revisited the paper draft again. Below is my thoughts after reading your rebuttal.
> >
> > - I insist that building upon pretrained model DINO is a severe concern. Please let me elaborate on this. I am not challenging recent works and recent progress for their use of DINO. I am well read with related works of LOST / Token Cut / Deep Spectral methods. Their claims are pretrained DINO features can enable accurate segmentations by some kinds of clustering/optimization, k-means or spectral methods. Importantly, they are not posing a new self-supervised pretext task or learning a new model from data. In your paper, this is very different. The claimed contribution of this paper is this new pretext task of synthesizing realistic images by shifting foreground objects. Given such an idea,  it is not obvious why it has to rely on a pretrained model especially with frozen features. In order to support the claim of the paper, I would expect the model to work with the segmentation network receiving raw RGB inputs. In your responses to other fellow reviewers, you mentioned that even fine-tuning the DINO model may not lead to satisfactory results. This actually shows that the pretext task is problematic for segmenting foreground objects.
> >
> > - I am not convinced that the pretext task relies on the claimed "movability of objects" for segmentation. I suspect that the method heavily relies on L_min, L_max, L_bin to approximate the size of the object heuristically. That being said, the region of the sky could certainly be shifted to synthesize realistic images. It is the contraints on the region size that force the model to segment object instead of the sky. The claim that object segmentations emerge from movability is weak.
> >
> > - On the writing quality on page 4, I apologize for being not concrete. The main approach contains a bulk of text from L87 to L114, which is disorganized.  To be specific, this single paragraph refers to Figure 1,2,3 respectively, and it is difficult for readers to grasp the overall pipeline of the approach. The paragraph starts by motivating inpainting and the use of MAE. However, it suddenly switches to the three losses L_min, L_max, L_bin, described earlier. This paragraph is also scrambled with high-level thoughts/motivation and implementation details. Moreover, I find the use of inline math equations with heavy notations to be hard to follow. I recommend to split this large paragraph into several manageable and organized paragraphs.
> >
> > - I am mostly clear on other concerns. Thanks.

---

> > > ### Author Response · Authors · 2022-08-03
> > > **Response to Reviewer tM6J - discussion**
> > >
> > > We thank the reviewer for continuing the discussion. Below is our response.
> > >
> > > - We’d like to thank the reviewer for very good suggestions to make the method description clearer. We’ll incorporate them in the next revision of the paper.
> > > - We understand the reviewer’s concern that our pretext task is not what drives the segmentation. However, we respectfully disagree and have data to support our claims.
> > > The validity of our reasoning is confirmed by the experiments in our ablation study. If the approach relied solely on the prior given by DINO’s (or any other model's) features and/or other mask constraints, we would be able to obtain meaningful segmentations without any shifts of the foreground. We show that those shifts are necessary to expose invalid segmentation masks. We show in Table 4 and section 4.3, lines 215-221 that this is the case; indeed, there is no incentive to produce any meaningful mask if we do not shift the foreground, but keep L_min, L_max and L_bin losses. In fact, in this case the segmenter trained on top of DINO features collapses to the smallest region satisfied by the L_min loss. The composite images in this case are indistinguishable from the inputs, but the masks are meaningless. Simply adding the local shifts of the foreground drives the mask into a meaningful solution.
> > > - We also show that DINO is not necessary for our approach, as we were able to train it with the features from MAE encoder as well. The study of fine-tuning of the entire DINO model will require much more work. The tuning of the ViT model in this case requires hyperparameter search, working with bigger datasets to avoid overfitting etc.
> > > - Even if MOVE needed pretrained features from a self-supervised model like DINO or MAE, we showed through ablation that our principles and especially the shifts are _necessary_ to make the model produce the meaningful masks. Compared to all the other methods that exploit DINO or other SSL features, ours gives SotA even without post training with pseudo ground truth masks.
> > > - It could be that a partial segmentation of the sky – or any uniform background – could provide a valid but incorrect segmentation. However, we can say that the vast majority of images does not have skies or uniform background, but they do have objects and that’s what the segmenter learns to pay attention to.

---

> > > > ### Comment · Reviewer_tM6J · 2022-08-04
> > > > **on the reliance of frozen pretrained features**
> > > >
> > > > I am afraid that the author responses failed to convince me.
> > > >
> > > > - The viable way to show this pretext task can learn meaningful object segmentations is to get ride of frozen pretrained features. Otherwise, object segmentations already emerge from DINO/MAE, and the technique of this paper is only to read it out, similar as k-means does. The excuse of extensive hyper-parameter search and working with bigger dataset is invalid. This is part of your job to be done to show the effectiveness of the pretext task.
> > > >
> > > > - Your ablation on the shifting operation cannot rebut my doubt. Without shifting, positives and negatives will be alike for GAN training. This no longer makes up a meaningful machine learning task. I can also understand that shifting is crucial for your method to work. However, this only shows that shifting is useful given frozen pretrained features, without having any evidence it can work beyond that.
> > > >
> > > > - Here is my understanding of your approach. Pretrained DINO (MAE similarly) features already enable well clustered embeddings for objects (shown by DINO, LOST). Your segmentation network is actually very small, comprising only 3 or 4 convolutional layers. In some sense, it is wrong to call this a segmentation network as it works like a head network. The shifting operation penalizes segmentations which go across the embedding boundaries. In my view, this paper proposes a clustering method given high quality features. There is no meaningful contributions to object segmentation and representation learning.
> > > >
> > > > Thanks.

---

> > > > > ### Author Response · Authors · 2022-08-04
> > > > > **Response**
> > > > >
> > > > > We disagree with the points raised by the reviewer.
> > > > >
> > > > > There is a contradiction in the argument, which we would like to point out.
> > > > >
> > > > > We will start with the most important claim made in the last statement, that is, whether MOVE provides objectively a meaningful contribution to object segmentation. The reviewer’s statement is that "In my view, this paper proposes a clustering method given high quality features. There is no meaningful contributions to object segmentation and representation learning." Also, in previous responses the following statement applied to prior published work in CVPR2022 "Their claims are pretrained DINO features can enable accurate segmentations by some kinds of clustering/optimization, k-means or spectral methods". Therefore, the statement is that our method is a clustering method as the other CVPR2022. However, in our case the reviewer says that there is "no meaningful contribution to object segmentation" despite our method achieving a significantly higher performance than prior work by using a very different approach.
> > > > >
> > > > > If the argument is that a pretrained network gives pretty much ready-to-use segmentation masks, then the statement is that all prior works are not meaningful contributions to object segmentation. However, the reviewers of those works disagree with this statement since these works were all accepted at CVPR22.
> > > > >
> > > > > If the argument is instead that using pretrained networks does not give ready-to-use segmentation masks and one needs a good criterion to extract valid segmentation masks, then the fact that prior work is valid, but our work is not, despite being also considered as a clustering method by the reviewer -- as all the other ones --, cannot be justified.
> > > > >
> > > > > In the following we argue why using pretrained networks is not a concern.
> > > > >
> > > > > 1) Pretrained networks such as DINO alone do not produce usable segmentation masks at the outset. Indeed, TokenCut, DeepSpectral etc need to introduce clustering criteria and handcrafted heuristics to obtain valid segmentation masks. Moreover, most methods use a re-training strategy based on the masks obtained through clustering because they are not sufficiently accurate. In contrast MOVE yields good masks immediately.
> > > > >
> > > > > 2) The principle we propose gives a useful bias towards the right segmentation. Any additional unsupervised bias, such as that from SSL pre-trained networks, is a valid support towards solving this really challenging task. So using pre-trained networks does not invalidate our method in any way.
> > > > >
> > > > > 3) Our ablation on the shifting operation does rebut the previous statement that "I am not convinced that the pretext task relies on the claimed "movability of objects" for segmentation. I suspect that the method heavily relies on L_min, L_max, L_bin to approximate the size of the object heuristically." The ablation shows that the other losses (L_min, L_max etc) are far from sufficient to define the segmentation masks. Moreover, by saying that without movability it "no longer makes up a meaningful machine learning task" the reviewer is saying that movability makes the difference and gives a meaningful machine learning task.
> > > > >
> > > > > Finally, the responses of the reviewer never address the fact that our proposed method yields a significant performance uniformly across all datasets used in prior evaluations.

---

> > ### Comment · Reviewer_tM6J · 2022-08-05
> > **fine-tuning does not work**
> >
> > I would expect the method to work from scratch for the best. I would be ok even if the method needs a pretrained network. However, it seems like it can only work with well-clustered features.
> >
> > Please answer my question: Does the failure to work for fine-tuning gives evidence that MOVE is not a good criterion for segmentation? As I said, it is not reasonable to excuse the hyper-parameters and etc. It is still a weakness that a method needs very dedicated hyper-parameter tuning.

---

> > > ### Author Response · Authors · 2022-08-05
> > > **Answers**
> > >
> > > The short answer to the question is no.
> > >
> > > The use of any pre-trained features that do not provide a ready-to-use segmentation mask, would not make the contribution of our method less significant.
> > > The evidence provided by the above mentioned CVPR22 works shows that DINO features require further clustering to provide valid segmentation masks.
> > > Therefore, the MOVE criterion must provide a non-trivial processing to produce valid segmentation masks.
> > > Our experiments give evidence that MOVE is a better criterion than methods in prior work.
> > > Therefore, the fine-tuning experiment does not invalidate the MOVE criterion. The evaluation of different pre-trained models and their fine-tuning is a separate study that, while interesting, is not necessary to demonstrate the MOVE criterion.
> > >
> > > Moreover, our method works also with not well-clustered features.
> > > We mentioned in our previous reply that we have successfully trained our segmenter also on top of a MAE Encoder features. MAE features are not known for being well-clustered. We have also visually inspected the correlation maps of MAE features for several image samples and compared them to those of DINO features. We can qualitatively conclude that they do not show a clear clustering.

---

> > > > ### Comment · Reviewer_tM6J · 2022-08-05
> > > > **thanks for your responses**
> > > >
> > > > I hope that I have given opportunities for the authors to address my concerns. It is unfortunate that I am not convinced by any of the arguments. I insist to keep my judgment unless full evidence is provided for fine-tuning or learning from scratch settings.
> > > >
> > > > Below, I briefly respond to your previous message. Prior works show that clustering methods as simple as k-means /spectral could work for segmentations upon Dino features. It does not have to be highly non-trivial processing. In fact, what MOVE learns is just 3 convolutional filters. To be honest, I have hand-on experience on playing with MAE features for clustering. A simple k-means would deliver reasonably good segmentations. MAE does learn well-clustered features. Since no concrete numbers and qualitative examples are provided for MAE throughout the paper (and rebuttal), I will not take this into account for my final evaluation.

---

> ### Author Response · Authors · 2022-08-08
> **Summary**
>
> Of course, we can provide numbers for the experiment with MAE encoder features. Below we present a comparison with DINO features for different datasets:
> | Dataset           | DINO features | MAE features |
> | ---                   |---                     |---                   |
> | ECSSD           | 0.809               | 0.795              |
> | DUTS-TE        | 0.680               | 0.646              |
> | DUT-OMRON | 0.585                | 0.575             |
>
> We’d like to clarify on the MAE features being well-clustered. The model that we used and which features we analyzed was the MAE model trained with a GAN loss. The features for this model are not well-clustered. The model the reviewer is probably referring to is a standard MAE model trained with a MSE loss only. Indeed, for this model we observe the features correlation map or k-means are showing some meaningful clustering. But this is not the model that we used. These MAE features yield a segmentation performance that is comparable to that of DINO features despite not being well-clustered.
>
> We might be unable to change the reviewer’s conviction about our work. Nonetheless, we’d like to end with a summary of points we made about our work:
> - Our method makes use of pre-trained self-supervised networks
> - MOVE neither depends on the exact choice of the inpainter nor the features
> - Our ablations show that it’s our learning task based on movability that enables segmentation
> - Our method achieves state-of-the-art results for unsupervised saliency segmentation, unsupervised single object discovery and unsupervised class-agnostic object detection
> - MOVE surpasses methods based on clustering of DINO features
>
> We have appreciated the discussion.

---

### Official Review · Reviewer_9LXw · 2022-07-12

**Rating:** 8
**Confidence:** 5
**Soundness:** 4 excellent
**Presentation:** 4 excellent
**Contribution:** 4 excellent

**Summary:**

This paper proposes a way to segment objects in unsupervised fashion.  The idea is to define a signal to a segmentation model based on the idea that if the object were properly segmented, then the predicted mask could be used to shift the object (by a small translation within the same image) in such a way that this new shifted image looks realistic as judged by a discriminator.  As part of generating this new shifted image of the object, the authors propose to use MAE (masked auto-encoders) to inpaint the missing region which is uncovered by the simulated shift.

The paper demonstrates that this proposed model “MOVE” outperforms comparable SOTA models on unsupervised saliency detection and unsupervised object detection benchmarks.


**Questions:**

* The cut+paste citation (Remez et al 2018) is slightly mischaracterized — this paper did not require pasting objects to different images (it was always to the same image).  Related to this, the Remez et al approach had an issue of inferring that shadows were part of an object.  What is the behavior of MOVE when there are shadows?
* In Section 5, the authors claim that MOVE does not require training of a generative model —- but this seems somewhat strange — is there a clear line that separates a model such as MAE from generative models?
* How necessary was it to use a DINO pretrained backbone?  Would starting from scratch have worked?


**Limitations:**

Yes

**Strengths And Weaknesses:**

I enjoyed reading this paper — the results are strong and seem to quite significantly outperform the state of the art.  Given the complexity of the approach, getting this level of performance is quite impressive and points to good execution.  Conceptually, the trick that the authors propose to deal with artifacts generated by MAE (specifically they propose to synthesize these artifacts in images where the object is unshifted which forces model to focus on differences other than the artifacts) is also simple and effective.

The main weakness of the approach is probably the complexity of the model which requires a segmentation model, a separately pre-trained masked auto-encoder and training both in a GAN-like setting, so I suspect that reproducing this model is not easy.

---

> ### Author Response · Authors · 2022-08-02
> **Response to Reviewer 9LXw**
>
> We thank the reviewer for the valuable feedback and useful comments.
>
> Below we'd like to answer the questions raised by the reviewer.
>
> -   *The main weakness of the approach is probably the complexity of the model which requires a segmentation model, a separately pre-trained masked auto-encoder and training both in a GAN-like setting, so I suspect that reproducing this model is not easy.*
>
>     -   We will provide the code for our paper, which will ensure model reproducibility. The other pre-trained models are readily available. We believe that this method is not more complex than most other methods for challenging tasks such as object detection. Notice that, although there are several components, there are only 2 trainable components: the segmenter and the discriminator. Moreover, we find that the training is stable on multiple datasets. We also find that our method is robust to the choice of the pre-trained models. In fact, we are also able to obtain comparable results with a single pre-trained model, by using MAE encoder features for segmentation instead of those from DINO. Similarly, our method works also when MAE is trained with an L2 loss (instead of a L2+GAN loss).
>
> -   *The cut+paste citation (Remez et al 2018) is slightly mischaracterized --- this paper did not require pasting objects to different images (it was always to the same image). Related to this, the Remez et al approach had an issue of inferring that shadows were part of an object. What is the behavior of MOVE when there are shadows?*
>
>     -   Thanks for pointing out the mischaracterization. We will amend the statement.
>
>         We notice that MOVE usually does not segment shadows together with a segmented object, but it often includes the object's reflection (e.g., a bird on a lake). Handling shadows is however a very interesting direction.
>
>
> -   *In Section 5, the authors claim that MOVE does not require training of a generative model ---- but this seems somewhat strange --- is there a clear line that separates a model such as MAE from generative models?*
>
>     -   We define generative models as those that map iid samples from a known noise distribution, eg, a Gaussian, to samples of data of interest (eg, images). Typically, the objective is for the generative model to produce new samples that belong to a distribution specified through a dataset. In contrast, our model (and other models such as MAE), map images to other images. The main difference is in the input and the objective of the training. In our experience, training generative models is much more challenging than mapping images to segmentation masks or performing inpainting.
>
>
> -   *How necessary was it to use a DINO pretrained backbone? Would starting from scratch have worked?*
>
>     -   We were able to substitute DINO features with features from the MAE encoder (which are also used for inpainting) and successfully train MOVE with this change, at the cost of a slightly decreased segmentation performance --- to be fair, the results are not directly comparable since MAE features have lower resolution (they encode 16x16 patches instead of 8x8 used by DINO), so the segmenter also needed to be adapted for that change. This suggests that DINO itself is not necessary, although a great initialization, and other self-supervised models can be used for feature extraction.
>     -   The model failed to produce anything meaningful when we tried to fine-tune the entire DINO ViT-S-8 network. For the smaller datasets, that we used for training, it is beneficial to use pre-trained self-supervised features. Nonetheless, we believe that the entire segmentation model could be trained from scratch. However, it would be computationally demanding (eg, require a resource-heavy architecture, hyperparameter search and a longer training schedule).

---

### Meta-Review · Area_Chair_Q8VZ · 2022-08-23

**Recommendation:** Accept
**Confidence:** Certain

**Metareview:**

This paper received mixed scores, with two reviewers recommending acceptance (Strong Accept) and one rejection. The paper was thoroughly discussed by the reviewers and the authors, but the reviewers failed to reach a consensus. Ultimately, after the authors' feedback having addressed most of the reviewers' concerns, the main point of disagreement that remains is the fact that the proposed method cannot be used to train or fine-tune the backbone parameters. While tM6J sees this as evidenced that the method is not sound, 9LXw and tSAq both see sufficient merit in the approach to consider this only as a minor drawback that will eventually be addressed in the future. Considering that the method nonetheless learns some parameters and produces convincing results, the AC agrees with 9LXw and tSAq that one paper does not necessarily need to address all problems. We nonetheless strongly encourage the authors to make this clear in the final version of the paper.

**Award:**

No

---

### Decision · Program_Chairs · 2022-09-14

Accept